# Diffusion Model for Graph Inverse Problems: Towards Effective Source Localization on Complex Networks

**Xin Yan**[1,*] **Hui Fang**[2,*] **Qiang He**[1,†]

[1]College of Medicine and Biological Information Engineering, Northeastern University, China
[2]RIIS & SIME, Shanghai University of Finance and Economics, China
yanx_in@outlook.com,fang.hui@mail.shufe.edu.cn,heqiangcai@gmail.com

## Abstract

Information diffusion problems, such as the spread of epidemics or rumors, are widespread in society. The inverse problems of graph diffusion, which involve locating the sources and identifying the paths of diffusion based on currently observed diffusion graphs, are crucial to controlling the spread of information. The problem of localizing the source of diffusion is highly ill-posed, presenting a major obstacle in accurately assessing the uncertainty involved. Besides, while comprehending how information diffuses through a graph is crucial, there is a scarcity of research on reconstructing the paths of information propagation. To tackle these challenges, we propose a probabilistic model called DDMSL (Discrete Diffusion Model for Source Localization). Our approach is based on the natural diffusion process of information propagation over complex networks, which can be formulated using a message-passing function. First, we model the forward diffusion of information using Markov chains. Then, we design a reversible residual network to construct a denoising-diffusion model in discrete space for both source localization and reconstruction of information diffusion paths. We provide rigorous theoretical guarantees for DDMSL and demonstrate its effectiveness through extensive experiments on five real-world datasets.

## 1 Introduction

Information diffusion is a pervasive phenomenon in our daily lives. Data scientists have conducted extensive research on information diffusion along the direction of entropy increase, such as maximizing the influence of nodes in social networks [15, 3], and developing control policies to curb the scale of epidemics in disease transmission networks [53]. However, merely comprehending the mechanism of forward diffusion is insufficient. When destructive information spreads across a network, it can cause huge damage on the entire system. For example, the rapid spread of rumors in social networks can cause harm to society [54], and the spread of computer viruses on the Internet can paralyze a system consisting of millions of users [1]. Additionally, pandemics such as SARS and COVID-19 in human interaction networks pose serious challenges to human health and social functioning [31]. Therefore, accurately identifying the sources of transmission and cutting off their possible transmission paths in time is crucial. This can help limit the spread of negative information and maintain the stability of the network.

Common information dissemination models like SIR (Susceptible-Infected-Recovered) and SI (Susceptible-Infected) [19] are subject to uncertainties during the process of information diffusion. As illustrated in Figure 1, the diffusion graph of information should follow a non-explicit distribution, implying that a single diffusion source may correspond to multiple different diffusion

---

*Equal contribution. †Corresponding author.

graphs, and a diffusion graph may have multiple different diffusion sources. Therefore, the inverse problem of information diffusion is underdetermined.

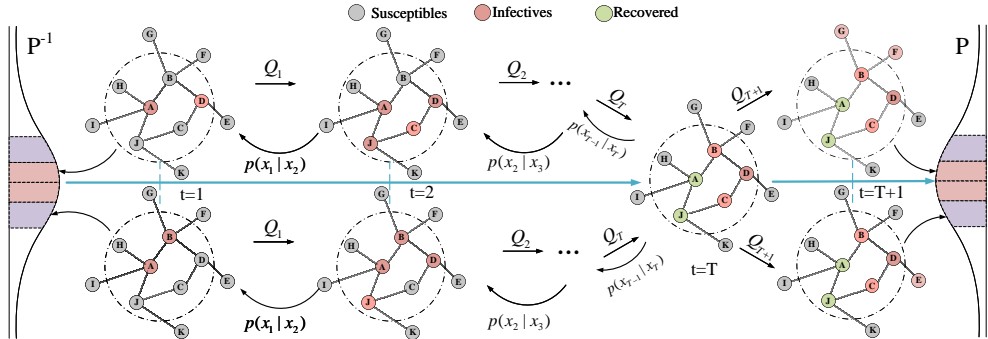

Figure 1: The SIR diffusion process of information over complex networks. Distinct sources may produce identical infection graphs ($x_T$), and initiating forward diffusion from $x_T$ may lead to different diffusion outcomes. The distribution of sources and forward diffusion can be represented by probability models $P^{-1}$ and $P$, respectively.

Previous source localization research has primarily relied on manually formulating rules to filter source nodes, such as using Jordan centrality [24, 26] to locate multiple source nodes in SI model, or unbiased intermediary centrality to identify source nodes in SIR model [49]. Besides, IVGD [43] exploited fixed point theorem for source localization of IC model [18]. However, these algorithms impose **relatively strict requirements on information diffusion patterns**, which could hardly be satisfied in real-world applications. To address this issue, some algorithms [38, 36] have been proposed using graph neural networks (GNN) [35] to learn source nodes under different diffusion models. Unfortunately, most deep learning-based algorithms ignore the underdeterminacy of information propagation and attempt to establish a direct mapping between observed data and source nodes. As a result, inference results are often deterministic and **fail to quantify the uncertainty of source localization**. In addition to source localization, **it is equally important to recover the possible propagation paths of information**. For example, when an infectious disease breaks out, it is necessary to promptly trace the virus transmission trajectory and corresponding close contacts. However, in reality, this can only be achieved by analyzing the genetic evolution tree of the virus strain [40], which is both time-consuming and labor-intensive. In summary, **there is a current dearth of methods that can simultaneously locate information sources and restore information propagation paths**.

Inspired by diffusion phenomena, the diffusion denoising model [16] has garnered significant achievements in the realm of image generation. It effectively restores the original image within a noise distribution through gradual refinement. This process bears resemblance to the challenges encountered in information reconstruction and diffusion, demanding our attention. Throughout the propagation process, the source information continually undergoes blurring, presenting an exceedingly arduous task of restoring the original source node. In this scenario, employing a diffusion model for the reverse recovery of the entire information dissemination process proves highly appropriate. However, using diffusion models to solve inverse problems in graphs presents several challenges: (1) Information diffusion occurs on non-Euclidean space graphs, making learning discrete non-Euclidean data difficult; (2) Diffusion over graphs is governed by information propagation rules that operate in non-Euclidean spaces, making it challenging to establish both the forward process and reverse inference process of information diffusion within the network; and (3) Most existing algorithms only work for specific propagation patterns, suffering from the problem of model generalization.

To address the aforementioned challenges, we propose a new framework, DDMSL[2](Discrete Diffusion Model for Source Localization). DDMSL excels at source localization and reconstructing the evolution of diffusion, showing promising results across different propagation patterns. Our contributions can be summarized as follows:

---

[2]The code for DDMSL is available at `https://github.com/Ashenone2/DDMSL`.

- We model the information diffusion process using Markov chains and demonstrate that at each moment, the state transition matrix converges, enabling us to reconstruct the information diffusion paths from any time point.

- We design a residual network block based on graph convolutional networks to approximate the information diffusion process and provide a theoretical proof for the model's reversibility.

- We propose an end-to-end framework being capable of both reconstructing the evolution of information diffusion and source localization. To the best of our knowledge, this is the first study to employ denoising diffusion models for solving graph inverse problems. Extensive experiments on five real-world datasets demonstrate its effectiveness.

## 2 Related work

**Diffusion models for inverse problems**. The diffusion denoising model [16, 29] is currently one of the best available deep generative models, which has been extensively applied in image-related inverse problems [33, 23, 34]. Previous studies have extended DP3Ms [2] to discrete spaces and used this work as a foundation for developing multiple generative diffusion models for graph data [14, 42]. Additionally, Stevens et al. [39] and Chung et al. [7] demonstrated that the diffusion model can solve nonlinear inverse problems. These works collectively demonstrate the effectiveness of diffusion models across different domains in addressing inverse problems.

**Localization of information sources on the graph**. To facilitate the localization of infection sources, various algorithms have been proposed [25, 52, 44, 9]. Early algorithms focused on screening source nodes by employing feature engineering. For instance, Luo et al. [25] identified the diffusion source by the centrality of the diffusion subgraphs, while Prakash et al. [30] and Zhu et al. [52] respectively developed NETSLEUTH and OJC algorithms based on the minimum description length of nodes and the shortest path. Wang et al. [44] designed a series of label propagation algorithms based on the aggregation process of information diffusion, on which Dong et al. [9] further improved using GCN [20]. Recently, the proposed reversible perception algorithm IVGD based on graph diffusion [43] has been applied to the IC model. However, **these algorithms generally have strict limitations on diffusion patterns or network structures.** DDMIX [8] was the first algorithm to use a generative model for reconstructing the dissemination paths of information. It leverages a VAE (variational autoencoder) to learn the state of nodes across all time steps; however, as the number of time steps increases, the solution space of the dissemination path becomes significantly more complex, leading to low accuracy in source localization. Ling et al. later introduced SLVAE [22], which is also based on VAE and can efficiently localize the source node but cannot directly reconstruct the propagation paths. Current generative model-based approaches for source localization **cannot simultaneously handle the problems of source localization and reconstructing diffusion paths**.

## 3 Methodology

### 3.1 Problem definition

We consider an undirected graph $\mathcal{G} = (\mathbf{V}, \mathbf{E})$ where $\mathbf{V} = \{x^1, x^2, \ldots, x^N\}$ is the node set and $\mathbf{E}$ is the edge set. Let $\mathbb{S} = \{x_0^{s_1}, x_0^{s_2}, \ldots x_0^{s_m}\}$ denote the set of initial diffusion source nodes at $t = 0$. The source nodes undergo diffusion on the graph $\mathcal{G}$ for $T$ time steps, governed by the diffusion pattern $g(\cdot)$, and the set of node states at time step $t$ is denoted by $\mathbf{X_t} = \{x_t^1, \ldots, x_t^N\}$ with $x_t^i \in \{0, 1\}^M$ (e.g., if $g(\cdot)$ is the SIR model, then $M \in \{S, I, R\}$, where $S, I, R$ denote susceptible, infected and recovered state, respectively). If the node $i \in \mathbb{S}$, then $x_0^i = 1$, otherwise $x_0^i = 0$. We define the research problem as finding the intermediate state $\mathbf{X} = \{\mathbf{X_0}, \mathbf{X_1}, \ldots, \mathbf{X_{T-1}}\}$ that maximizes the likelihood function $\mathbf{X}^* = argmax_{\mathbf{X}}^{P\{\mathbf{X_T}|\mathbf{X}, \mathcal{G}\}}$, given the observed $\mathbf{X_T}$.

### 3.2 Information diffusion in discrete spaces

Susceptible-Infected-Recovered (SIR) and Susceptible-Infected (SI) models [19] are commonly used to model diffusion phenomena in nature, such as the spread of epidemics [51] and rumors [50]. In this paper, we will demonstrate our approach using the SIR model. The SIR model categorizes node states into susceptible (S), infected (I), and recovered (R) states. The S state represents individuals who are

susceptible to infection, while the I state represents those who have been infected, and the R state indicates recovery from the infected state. The transition between these three states is irreversible. We assume that all nodes on the graph are homogeneous and follow the same transition process for the different states of the SIR model[3]:

$$
\begin{cases}
P\left(x_{t+1}^i = I \mid x_t^i = S\right) = 1 - \prod_j \left(1 - \beta_I^j(t) A_{ij} I_j(t)\right) \\
P\left(x_{t+1}^i = R \mid x_t^i = I\right) = \gamma_R^i(t)
\end{cases}
\tag{1}
$$

And, the SIR diffusion process can be represented using a state transfer matrix:

$$
Q_t^i =
\begin{bmatrix}
1 - \beta_I^i(t) & \beta_I^i(t) & 0 \\
0 & 1 - \gamma_R^i(t) & \gamma_R^i(t) \\
0 & 0 & 1
\end{bmatrix}
\tag{2}
$$

where $Q_t^i$ denotes the state transfer matrix of node $i$ at moment $t$, and $\left[Q_t^i\right]_{uv}$ denotes the probability of transferring from state $u$ to state $v$[4]. $\beta_I^i(t)$ and $\gamma_R^i(t)$ denote the infection rate and recovery rate at moment $t$, respectively. Let $[P_S^i(t), P_I^i(t), P_R^i(t)]$ to be the probabilities of node $i$ being in three states $S$, $I$, and $R$ at time $t$, then $\beta_I^i(t)$ and $\gamma_R^i(t)$ can be calculated by the following equation:

$$
\begin{cases}
\beta_I^i(t) = \dfrac{P_I^i(t+1) - P_I^i(t)(1 - \gamma_R^i(t))}{P_S^i(t)} = 1 - \dfrac{P_S^i(t+1)}{P_S^i(t)} \\[2mm]
\gamma_R^i(t) = \dfrac{P_R^i(t+1) - P_R^i(t)}{P_I^i(t)}
\end{cases}
\tag{3}
$$

**Theorem 3.1** *Given graph $\mathcal{G}$ and the infected seed set $\mathbb{S}$, it can be determined that the state transfer matrix $Q_t^i$ for the $SIR$ diffusion process on $\mathcal{G}$ converges at all times.*

**Sketch of Proof**. Given $\mathcal{G}$ and $\mathbb{S}$, the initial state of information diffusion can be determined. Based on the SIR propagation rule, an iterative equation can be constructed for the state distribution of the nodes at any given time. Finally, $Q_t^i$ can be calculated according to Equations 2 and 3.

Referring to Equations 2 and 3, we can readily demonstrate the convergence of the state transition matrix $Q_t^i$ at every moment, rendering the discrete diffusion model a practicable solution [2]. The proof is shown in Theorem3.1.

### 3.3 Discrete diffusion model for source localization

#### 3.3.1 Forward process

We represent the state of node $i$ by an one-hot vector $\boldsymbol{x}_t^i \in \mathbb{R}^{1 \times M}$, and the state distribution of node $i$ at time $t$ is written as:

$$
q\left(\boldsymbol{x}_t^i \mid \boldsymbol{x}_{t-1}^i\right) = \boldsymbol{x}_{t-1}^i Q_t^i \boldsymbol{x_t^i}^T \sim \mathrm{Cat}\left(\boldsymbol{x}_t^i; \boldsymbol{p} = \boldsymbol{x}_{t-1}^i Q_t^i\right)
\tag{4}
$$

Information diffusion is a Markov process that allows for inference of the node state at any given moment based on the initial state:

$$
q\left(\boldsymbol{x}_t^i \mid \boldsymbol{x}_0^i\right) = \sum_{\boldsymbol{x}_{1:t-1}^i} \prod_{k=1}^t q\left(\boldsymbol{x}_k^i \mid \boldsymbol{x}_{k-1}^i\right) = \boldsymbol{x}_0^i \bar{Q}_t^i \boldsymbol{x_t^i}^T \sim \mathrm{Cat}\left(\boldsymbol{x}_t^i; \boldsymbol{p} = \boldsymbol{x}_0^i \bar{Q}_t^i\right)
\tag{5}
$$

#### 3.3.2 Reverse process

Reconstructing the information diffusion process requires backward inference of the forward process, which can be achieved through Bayesian formulation[5]:

$$
q\left(\boldsymbol{x}_{t-1}^i \mid \boldsymbol{x}_t^i, \boldsymbol{x}_0^i\right) \sim \mathrm{Cat}\left(\boldsymbol{x}_{t-1}^i; \boldsymbol{p} = \frac{\left(\boldsymbol{x}_t^i {Q_t^i}^T\right) \odot \left(\boldsymbol{x}_0^i \bar{Q}_{t-1}^i\right)}{\boldsymbol{x}_0^i \bar{Q}_t^i \boldsymbol{x_t^i}^T}\right)
\tag{6}
$$

---

[3]$A$ denotes the graph's adjacency matrix, and $I_j(t)$ represents whether neighbor $j$ is in an infected state at $t$.

[4]In the SIR model, $u, v \in \{1, 2, 3\}$, representing the three states: S, I and R, respectively.

[5]Please refer to Appendix B for the formula derivation of forward and backward processes.

In the absence of knowledge about $x_0^i$, the posterior distribution $q\left(x_{t-1}^i \mid x_t^i, x_0^i\right)$ becomes intractable to compute directly. As a result, we must approximate this distribution using other methods. Continuing with Bayes' theorem, we can express the posterior distribution as follows:

$$q\left(\boldsymbol{x}_{t-1}^i \mid \boldsymbol{x}_t^i, \boldsymbol{x}_0^i\right) = q\left(\boldsymbol{x}_{t-1}^i \mid \boldsymbol{x}_t^i\right) = \frac{\sum_{\boldsymbol{x}_0^i} q\left(\boldsymbol{x}_{t-1}^i, \boldsymbol{x}_t^i, \boldsymbol{x}_0^i\right)}{q\left(\boldsymbol{x}_t^i\right)} = \mathbb{E}_{q\left(\boldsymbol{x}_0^i \mid \boldsymbol{x}_t^i\right)} q\left(\boldsymbol{x}_{t-1}^i \mid \boldsymbol{x}_t^i, \boldsymbol{x}_0^i\right) \quad (7)$$

**We utilize a neural network model ($nn_\theta$) to learn about $p_\theta\left(x_{t-1}^i \mid x_t^i\right)$ and estimate $q\left(x_0^i \mid x_t^i\right)$. The detailed design of $nn_\theta$ will be elaborated later in Section 3.4.**

$$q\left(\boldsymbol{x}_{t-1}^i \mid \boldsymbol{x}_t^i\right) \approx p_\theta\left(\boldsymbol{x}_{t-1}^i \mid \boldsymbol{x}_t^i\right) = \mathbb{E}_{\boldsymbol{x}_0^i \sim p_\theta\left(\boldsymbol{x}_0^i \mid \boldsymbol{x}_t^i\right)} q\left(\boldsymbol{x}_{t-1}^i \mid \boldsymbol{x}_t^i, \boldsymbol{x}_0^i\right)$$

$$= \frac{q\left(\boldsymbol{x}_t^i \mid \boldsymbol{x}_{t-1}^i\right) \left[\sum_j q\left(\boldsymbol{x}_{t-1}^i \mid \boldsymbol{x_0^i}^{(j)}\right) p_\theta\left(\boldsymbol{x_0^i}^{(j)} \mid \boldsymbol{x}_t^i\right)\right]}{q\left(\boldsymbol{x}_t^i \mid \boldsymbol{x}_0^i\right)} \quad (8)$$

Similar to [2], we predict $x_0^i$ and then use this prediction to learn the distribution of $q\left(x_0^i \mid x_t^i\right)$. This approach offers several benefits. First, it leverages prior knowledge by exploiting the fact that at $t = 0$ there are only two states: susceptible and infected. This makes $nn_\theta$ easier to train. Second, locating $x_0^i$ is a key target of our task. Finally, predicting $x_0^i$ can help constrain errors in the information reconstruction process (Refer to Appendix C).

$$L_{\mathrm{vb}}^i = \mathbb{E}_{q\left(\boldsymbol{x}_0^i\right)} \left[ \underbrace{D_{\mathrm{KL}}\left[q\left(\boldsymbol{x}_T^i \mid \boldsymbol{x}_0^i\right) \| p\left(\boldsymbol{x}_T^i\right)\right]}_{L_T} + \right.$$

$$\left. \sum_{t=2}^T \underbrace{\mathbb{E}_{q\left(\boldsymbol{x}_t^i \mid \boldsymbol{x}_0^i\right)}[D_{\mathrm{KL}}\left[q\left(\boldsymbol{x}_{t-1}^i \mid \boldsymbol{x}_t^i, \boldsymbol{x}_0^i\right) \| p_\theta\left(\boldsymbol{x}_{t-1}^i \mid \boldsymbol{x}_t^i\right)\right]]}_{L_{t-1}} - \underbrace{\mathbb{E}_{q\left(\boldsymbol{x}_1^i \mid \boldsymbol{x}_0^i\right)}\left[\log p_\theta\left(\boldsymbol{x}_0^i \mid \boldsymbol{x}_1^i\right)\right]}_{L_0} \right]$$

$$(9)$$

Equation 9 presents the variational lower bound loss function of the denoising diffusion model [16], where $D_{KL}$ represents relative entropy. Furthermore, as highlighted in our earlier discussion, it is crucial for us to supervise $x_0^i$ directly. As a result, we arrive at the simplified variational lower bound loss ($L_{simple}$), which is denoted as:

$$L_{simple} = \frac{1}{N} \sum_{i=1}^N \left(L_{vb}^i + \mathbb{E}_{q\left(\boldsymbol{x}_0^i\right)} \mathbb{E}_{q\left(\boldsymbol{x}_t^i \mid \boldsymbol{x}_0^i\right)} \left[-\log \widetilde{p}_\theta\left(\boldsymbol{x}_0^i \mid \boldsymbol{x}_t^i\right)\right]\right) \quad (10)$$

### 3.3.3 Supervision of information propagation rules

Our reconstructed information diffusion must adhere to the propagation rules of model $g(\cdot)$. During early training, the predictions of $nn_\theta$ for $x_{t-1}^i$ may not be accurate enough, which could lead to violations of the propagation rules. For example, using the SIR model, if the state of node $i$ at time $t$ is classified as $I$, then the state of $i$ at time $t-1$ can only be either $S$ or $I$. However, the prediction result of $nn_\theta$ may erroneously output state $R$, resulting in the failure to calculate $q\left(x_{t-1}^i \mid x_t^i, x_0^i\right)$.

To prevent violations of the propagation rules, we can take two measures: (1) In cases where the inference result of $nn_\theta$ violates the propagation rule, $q(x_{t-1}^i \mid x_t^i, x_0^i)$ should be set to $[1, 0, 0]$. and (2) To supervise the training of $nn_\theta$, the propagation rule loss function $L_{constrain} = L_{constrain1} + L_{constrain2}$ is introduced. The detailed derivations of (1) and (2) are shown in Appendix C.

$$\begin{cases} L_{constrain1} = Relu\left(\mathbf{X_{t-1}} - (\mathbf{A} + \mathbf{I})\mathbf{X_0}\right) \\ L_{constrain2} = \left\|\max\left(\mathbf{0}, \mathbf{X_{t-1}^{(j)}} - \mathbf{X_{t-1}^{(i)}}\right)\right\|_2^2, \forall \mathbf{X_0^{(i)}} \supseteq \mathbf{X_0^{(j)}} \end{cases} \quad (11)$$

In order to evaluate Equation 11, we must first sample $\mathbf{X_{t-1}}$ from the discrete probability distribution computed in Equation 8. To ensure gradient preservation during the sampling process, we employ the $Gumbel\text{-}Softmax$ technique [17]: $\mathbf{X_{t-1}} \sim Gumbel\text{-}Softmax(q(x_{t-1}^i \mid x_t^i, x_0^i))$. Equation 11 demonstrates that each node's contribution to the information diffusion process is non-negative. In summary, the constrained loss function is $L_{constrain=} = L_{constrain1} + L_{constrain2}$, and the loss function of DDMSL is:

$$Loss = L_{simple} + L_{constrain} \quad (12)$$

## 3.4 Design of the inference model

**In this section, we discuss the design of the $nn_\theta$ in detail**. Let $\mathbf{X_0} = \{x_0^1, x_0^2 \ldots x_0^N\}$ and $\mathbf{Y_T} = \{x_T^1, x_T^2 \ldots, x_T^N\}$. The diffusion of information on complex networks can be modeled using the following processes [46, 43]:

$$\mathbf{X_0} \xrightarrow{f_w} \xi \xrightarrow{\mathbf{F}} \mathbf{Y_T} \qquad (13)$$

Within this framework, $f_w$ and $\mathbf{F}$ serve as the feature vector construction and feature propagation func-

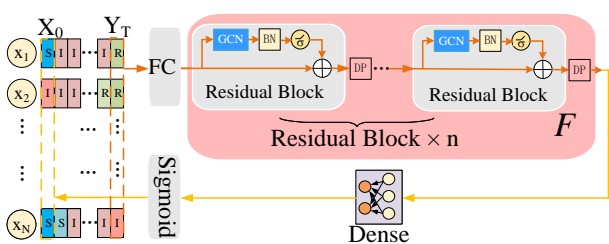

Figure 2: Multi-layer reversible residual network.

tions, respectively. The relationship between $\mathbf{Y_T}$ and $\mathbf{X_0}$ be expressed as: $\mathbf{Y_T} = \mathbf{P}(\mathbf{X_0}) = \mathbf{F}(f_w(\mathbf{X_0}))$. Our research objective focuses on the inverse problem of the SIR or SI diffusion model, which entails a large number of stochastic processes. Thus, when $\mathbf{F} = \mathbf{RD}$, solving $\mathbf{P}^{-1}$ can be extremely challenging. In the following subsections, we will explore how to construct suitable functions $f_w$ and $\mathbf{F}$ to facilitate the calculation of $\mathbf{P}^{-1}$.

### 3.4.1 Relationship between GCN and SIR diffusion models

According to [37], diffusion models such as SIR are analogous to message passing neural networks (MPNNs [11]), where each node's state is only updated based on the states of its neighboring nodes. However, the operational architecture must be designed to enable each node to aggregate both its own features and those of its neighbors, and update its state via nonlinear activation as shown in Equation 26. Therefore, we propose using MPNNs to remodel the SIR diffusion process. Denoting $P_\Omega^i(t) \equiv P(x_t^i = \Omega)$ as the probability that node $i$ is in state $\Omega \in \{S, I, R\}$ at time $t$ in the SIR diffusion process. It can be observed that this process is structurally equivalent to:

$$\begin{cases} m_\Omega^i(t+1) & = \sum_{j \in N(i)} M_t \left( h(P_\Omega^i(t)), h(P_\Omega^j(t)), e_{ij} \right) \\ P_\Omega^i(t+1) & = U_t \left( h(P_\Omega^i(t)), m_\Omega^i(t+1) \right) \\ h(P_\Omega^i(t)) & = \sigma \left( \mathbf{W_\Omega} P_\Omega^i(t) + \mathbf{b_\Omega} \right) \end{cases} \qquad (14)$$

where $M_t$ is the aggregate function and $U_t$ represents the node status update function. $\sigma(\cdot)$ is a nonlinear activation function. $e_{ij}$ represents the edge between nodes i and j. Additionally, since $h(P_\Omega^i(t))$ already contains nonlinear transformations, more complex forms of transformation are not necessary and $U_t$ can be defined as: $U_t(a, b) = a + b$. To sum up, Equation 14 can be simplified as:

$$P_\Omega^i(t+1) = h(P_\Omega^i(t)) + \sigma \left( \sum_{j \in N(i)} h(P_\Omega^j(t)) \right) \qquad (15)$$

We can achieve this using a residual block composed of graph convolutional networks, which allows us to easily fit the SIR model:

$$\begin{cases} \mathbf{h}_{i,t}^{(0)} = \mathbf{SN}(\mathbf{U} x_i^t) \text{ with } \mathbf{U} \in \mathbb{R}^{C \times M} \\ g(\mathbf{h}) = \sigma_g(\mathbf{D}^{-1/2} \mathbf{A} \mathbf{D}^{-1/2} \cdot \mathbf{h} \cdot \mathbf{w} + \mathbf{b}) \\ \mathbf{h}_{i,t}^{(l+1)} = \mathbf{h}_{i,t}^{(l)} + \sigma \left( \mathbf{SN} \left( \mathbf{BN} \left( g \left( \mathbf{h}_{i,t}^{(l)} \right) \right) \right) \right) \end{cases} \qquad (16)$$

Among them, $\mathbf{U}$ is a linear transformation, $\mathbf{h}_{i,t}^{(l)}$ denotes the representation of the $l$-th layer of the network, $\mathbf{D}$ is the degree matrix, $\mathbf{SN}(\cdot)$ and $\mathbf{BN}(\cdot)$ represent spectral normalization and batch normalization, respectively. $\sigma(\cdot)$ and $\sigma_g(\cdot)$ are nonlinear activation functions[6]. In fact, $f_w$ and $\mathbf{F}$[7] in Equation 13 are the $\mathbf{SN}(\mathbf{U}(\cdot))$ and residual blocks in Equation 16.

---

[6] We set $\sigma(\cdot)$ and $\sigma_g(\cdot)$ to $Mish$ and $LeakyReLu$ respectively.
[7] $\mathbf{F} = \xi + f_\theta(\xi)$, where $f_\theta(\xi) = \sigma(\mathbf{SN}(\mathbf{BN}(\xi)))$.

### 3.4.2 Reversibility of residual blocks

We now discuss the invertibility of Equation 13 as reconstructed from Equation 16. Equation 13 can be written as:

$$\begin{cases} \mathbf{SN}\left(\mathbf{U}(\mathbf{X_0})\right) = f_w(\mathbf{X_0}) = \xi; \\ \xi + \sigma\left(\mathbf{SN}\left(\mathbf{BN}\left(g(\xi)\right)\right)\right) = \xi + f_\theta(\xi) = \mathbf{F}(\xi) = \mathbf{Y_T}. \end{cases} \tag{17}$$

**Lemma 3.1** *Denoting the Lipschitz constants of $f_w$ and $f_\theta$ to be $L_w$ and $L_\theta$, then $L_w < 1$ and $L_\theta < 1$.*

**Sketch of Proof**. Since $f_w$ and $\mathbf{BN}\left(g(\xi)\right)$ are spectrally normalized, their Lipschitz coefficients are less than 1 [28]. Additionally, since the $\sigma(\cdot)$ is set to $Mish(\cdot)$, it can be determined that the Lipschitz constant of their composite function (i.e., $f_\theta$) is less than 1.

**Theorem 3.2** *If $L_w < 1$ and $L_\theta < 1$, then $\mathbf{Y_T} = \mathbf{F}(f_w(\mathbf{X_0}))$ is reversible.*

**Sketch of Proof**. $\mathbf{Y_T} = \mathbf{F}(f_w(\mathbf{X_0}))$ can be written as $\begin{cases} \mathbf{X_0} = \xi + \mathbf{X_0} - f_w(\mathbf{X_0}) \\ \xi = \mathbf{Y_T} - f_\theta(\xi) \end{cases}$. Since $f_w < 1$ and $f_\theta < 1$, constructing the iteration according to the Banach fixed point theorem [4], $\mathbf{Y_T} = \mathbf{F}(f_w(\mathbf{X_0}))$ is invertible.

In practical applications, we typically use multiple residual blocks to form a residual network. This approach is employed to improve the receptive field of the GCN and alleviate the problem of node smoothing caused by multi-layer graph convolution. The multi-layer reversible residual network that we designed is depicted in Figure 2, where $\mathbf{DP}$ represents dropout. If the dropout rate is $r$ and the $i$-th layer residual block is $\mathbf{F_i}$, then we have $\mathbf{F_i}(\xi) = \mathbf{DP}(\xi + f_w(\xi))$.

**Theorem 3.3** *If $dropout\text{-}rate = 0.5$, $L_w < 1$, and $L_\theta < 1$, the residual network $\mathbf{Y_T} = (\mathbf{F_1} \circ \mathbf{F_2} \circ \ldots \circ \mathbf{F_n})(f_w(\mathbf{X_0}))$ is reversible.*

**Sketch of Proof**. Denoting this multilayer residual network as $F$, the upper bound of $L_F$ is the product of the Lipschitz constants of each function [13]. Additionally, with a dropout rate of $r$, the Lipschitz constants of the functions will be limited to $(1 - r)$ times their original values. Using these two conclusions, we can calculate the upper bound of the Lipschitz constant of a multilayer reversible network with $L_F \leq 1$ when $r = 0.5$.

## 4 Experiments

### 4.1 Datasets and evaluation metrics

**Datasets**. The diffusion of information occurs in a broad range of network types, and DDMSL was evaluated on five distinct realistic datasets: **Karate** [48], **Jazz** [12], **Cora-ML** [27], **Power Grid** [45], and **PGP** [5]. The detailed parameters of these networks are provided in the Appendix F.2. Following previous research [43, 9, 22], we conducted SIR and SI diffusion simulations on each dataset by randomly selecting 10% of the nodes as source nodes at the initial moment, and stopping the simulation when approximately 50% of the nodes were infected[8] (For PGP dataset, simulation stopped at 30% infection rate). We randomly divided each generated dataset into a training set, a validation set and a test set in the ratio of $8 : 1 : 1$.

**Evaluation Metrics**. Our objective consists of two components: source localization and information diffusion paths recovery. The source localization task is a binary classification task, and thus evaluated using four metrics: Precision (**PR**), which denotes the proportion of nodes predicted as sources that are true sources; Recall (**RE**), which represents the proportion of actual source nodes that are correctly predicted; **F1** score, the harmonic mean of PR and RE; and ROC-AUC (**AUC**), quantifying the model's ability to classify accurately. To evaluate the performance of the recovering information diffusion paths, we adopted the Mean Squared Error (MSE) error in [8]: $MSE = \frac{1}{NT} \sum_{t=0}^{T-1} \left\| \mathbf{X_t} - \hat{\mathbf{X}_t} \right\|^2$, where $\hat{X}_t$ is the ground truth. This metric is solely computed for infected nodes, with nodes in the recovered and susceptible states being marked as 0.

---

[8]While some previous studies [9, 44] commenced source localization when the infection rate reached 30%, we set it to 50% to consider more complex scenarios.

## 4.2 Baseline algorithms and experimental settings

Six algorithms for source location were selected to compare with DDMSL: **DDMIX** [8] employs a Variational Autoencoder (VAE) to reconstruct the outbreak propagation process; **LPSI** [44] aggregates information to learn source detection through label propagation; **GCNSI** [9] uses graph convolutional network to learn manually formulated node features, enabling the discrimination of multiple source nodes; **OJC** [52] locates the source node by minimising the eccentricity of the infection; **NETSLEUTH** [30] filters multi-source nodes based on minimum description distance, but works solely on the SI diffusion model; and **SLVAE** [22] deploys generative models to learn the distribution of source nodes. Appendix F.1 provides the experimental settings and tuned hyperparameters for these algorithms, while implementation details on DDMSL are shown in Appendix E.

## 4.3 Experimental results

while comparing DDMSL with other algorithms on different diffusion models, we have focused on the widely used SIR and SI models, which are effective in capturing the dynamics of various information diffusion processes found in nature. It is noteworthy that DDMSL's flexibility and robustness enable it to handle real-world scenarios where the diffusion process may undergo significant variations by simply designing the corresponding state transfer matrices[9].

Table 1: Performance comparison (mean of five rounds) of source localization under the SIR model. The best results are highlighted in bold, while the underlined indicates the best of the five baselines. Statistical significance from paired t-test depicts: $***$, $**$, $*$ for p-value $< 0.01, 0.05, 0.1$ respectively.

| Methods | Karate | | | | Jazz | | | | Cora Ml | | | | Power Grid | | | | PGP | | | |
|---|---|---|---|---|---|---|---|---|---|---|---|---|---|---|---|---|---|---|---|---|
| | PR | RE | F1 | AUC | PR | RE | F1 | AUC | PR | RE | F1 | AUC | PR | RE | F1 | AUC | PR | RE | F1 | AUC |
| DDMSL | **0.708** | **0.736** | **0.722** | **0.853** | **0.817** | **0.881** | **0.848** | **0.930** | **0.894** | **0.867** | **0.880** | **0.928** | 0.833 | **0.879** | **0.855** | **0.930** | **0.856** | **0.903** | **0.879** | **0.943** |
| GCNSI | 0.275 | 0.410 | 0.329 | 0.671 | 0.301 | 0.363 | 0.330 | 0.641 | 0.247 | 0.273 | 0.260 | 0.591 | 0.165 | 0.182 | 0.173 | 0.540 | 0.554 | 0.543 | 0.549 | 0.748 |
| LPSI | 0.211 | 0.393 | 0.274 | 0.646 | 0.400 | 0.098 | 0.158 | 0.543 | 0.246 | 0.026 | 0.048 | 0.509 | 0.193 | 0.012 | 0.022 | 0.503 | 0.518 | 0.437 | 0.474 | 0.696 |
| SLVAE | 0.552 | 0.400 | 0.464 | 0.696 | 0.750 | 0.576 | 0.651 | 0.778 | 0.815 | 0.721 | 0.765 | 0.852 | **0.908** | 0.719 | 0.803 | 0.856 | 0.817 | 0.721 | 0.766 | 0.851 |
| OJC | 0.178 | 0.265 | 0.213 | 0.594 | 0.147 | 0.161 | 0.154 | 0.535 | 0.114 | 0.114 | 0.114 | 0.508 | 0.109 | 0.109 | 0.109 | 0.505 | 0.128 | 0.128 | 0.128 | 0.516 |
| DDMIX | 0.289 | 0.234 | 0.258 | 0.308 | 0.215 | 0.197 | 0.205 | 0.238 | 0.162 | 0.273 | 0.204 | 0.250 | 0.333 | 0.253 | 0.287 | 0.346 | 0.172 | 0.194 | 0.182 | 0.213 |
| Improve. | 28.3% | 84.3% | 55.7% | 22.5% | 8.9% | 52.9% | 30.1% | 19.6% | 9.7% | 20.2% | 15.0% | 8.9% | -8.3% | 22.2% | 6.6% | 8.7% | 4.8% | 25.3% | 14.8% | 10.8% |
| Significance | *** | *** | *** | *** | ** | *** | ** | * | *** | * | ** | * | *** | *** | *** | *** | *** | *** | *** | *** |

Table 2: Performance comparison (mean of five rounds) of source localization under the SI model. The best results are highlighted in bold, while the underlined indicates the best of the six baselines. Statistical significance from paired t-test depicts: $***$, $**$, $*$ for p-value $< 0.01, 0.05, 0.1$ respectively.

| Methods | Karate | | | | Jazz | | | | Cora Ml | | | | Power Grid | | | | PGP | | | |
|---|---|---|---|---|---|---|---|---|---|---|---|---|---|---|---|---|---|---|---|---|
| | PR | RE | F1 | AUC | PR | RE | F1 | AUC | PR | RE | F1 | AUC | PR | RE | F1 | AUC | PR | RE | F1 | AUC |
| DDMSL | **0.706** | **0.980** | **0.798** | **0.972** | 0.782 | **0.853** | **0.813** | **0.914** | 0.790 | **0.908** | **0.845** | **0.941** | 0.763 | **0.966** | **0.852** | **0.966** | 0.754 | **0.887** | **0.815** | **0.928** |
| GCNSI | 0.357 | 0.456 | 0.401 | 0.687 | 0.366 | 0.426 | 0.394 | 0.676 | 0.321 | 0.354 | 0.337 | 0.636 | 0.335 | 0.325 | 0.330 | 0.639 | 0.487 | 0.370 | 0.421 | 0.665 |
| LPSI | 0.339 | 0.414 | 0.351 | 0.681 | 0.474 | 0.097 | 0.156 | 0.544 | 0.494 | 0.207 | 0.291 | 0.592 | 0.343 | 0.277 | 0.306 | 0.609 | 0.453 | 0.284 | 0.349 | 0.623 |
| SLVAE | 0.591 | 0.477 | 0.503 | 0.733 | **0.888** | 0.579 | 0.691 | 0.785 | **0.841** | 0.728 | 0.780 | 0.856 | **0.815** | 0.780 | 0.797 | 0.880 | **0.844** | 0.633 | 0.723 | 0.810 |
| OJC | 0.267 | 0.396 | 0.318 | 0.663 | 0.120 | 0.127 | 0.123 | 0.517 | 0.125 | 0.125 | 0.125 | 0.514 | 0.178 | 0.178 | 0.178 | 0.544 | 0.118 | 0.118 | 0.118 | 0.510 |
| DDMIX | 0.253 | 0.377 | 0.303 | 0.654 | 0.244 | 0.133 | 0.172 | 0.212 | 0.220 | 0.222 | 0.221 | 0.247 | 0.345 | 0.235 | 0.280 | 0.340 | 0.189 | 0.186 | 0.187 | 0.207 |
| NetSleuth | 0.239 | 0.339 | 0.279 | 0.634 | 0.216 | 0.252 | 0.233 | 0.580 | 0.217 | 0.229 | 0.223 | 0.569 | 0.206 | 0.216 | 0.211 | 0.562 | 0.200 | 0.210 | 0.205 | 0.558 |
| Improve. | 19.4% | 105.5% | 58.5% | 32.5% | -11.9% | 47.3% | 17.6% | 16.3% | -6.0% | 24.8% | 8.3% | 9.9% | -6.3% | 23.8% | 7.0% | 9.8% | -10.6% | 40.2% | 12.7% | 14.5% |
| Significance | *** | *** | *** | *** | *** | *** | *** | *** | *** | *** | *** | *** | * | * | ** | ** | *** | *** | *** | *** |

**Source Location**. Table 1 shows the performance of DDMSL in SIR diffusion mode, where it outperforms all baseline algorithms regarding almost all metrics. Notably, DDMSL leads the best of the baseline algorithms by approximately 25% and 14% on F1 score and AUC, respectively. Meanwhile, Table 2 shows similar results in the SI diffusion mode. While SLVAE has a higher precision than DDMSL, DDMSL surpasses SLVAE by approximately 21% and 17% in terms of F1 score and AUC, indicating that DDMSL is more accurate in identifying source nodes. Moreover, the performance of DDMSL varies across datasets and is mainly influenced by the network's typologies (Refer to Table 6) and the scale of information diffusion. The performance of DDMSL in solving the inverse problem generally has a negative correlation with the average clustering coefficient and density of the network. The larger these two parameters are, the more complex the scenario of information diffusion is, and the accuracy of DDMSL in detecting source points decreases, which aligns with our intuition.

---

[9]Please refer to Appendix E for measuring the state transition matrices for different propagation models.

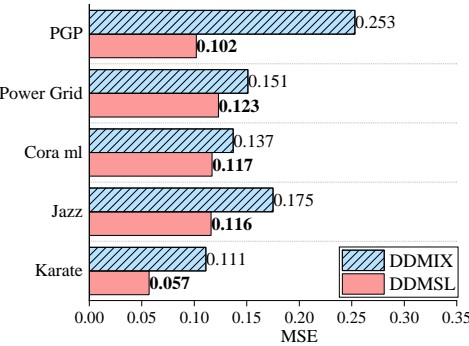

(a) MSE reconstruction error in SIR mode.

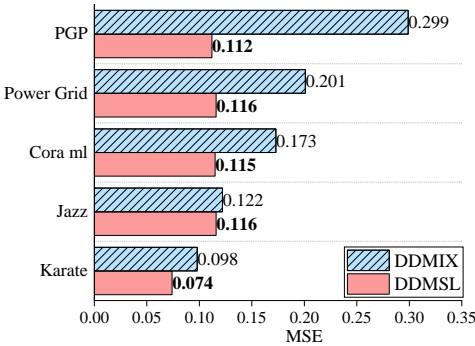

(b) MSE reconstruction error in SI mode.

Figure 3: Comparisons of DDMSL and DDMIX on reconstructed information diffusion processes.

**Reconstructing information diffusion paths**. Both DDMSL and DDMIX can reconstruct the evolution of information diffusion, but DDMIX can only recover the states of susceptible nodes ($S$) and infected nodes ($I$), whereas DDMSL can reconstruct nodes of all states. To facilitate fair comparison, we calculate MSE loss for node reconstruction in states $S$ and $I$, as shown in Figure 3. DDMSL consistently outperforms DDMIX on all datasets, especially in SI diffusion mode where DDMSL's average reconstruction loss is 69% lower than DDMIX's. Unlike DDMIX, which recovers the entire graph state, DDMSL allows for fine-grained node-level reconstruction.

**Additional experiments**. DDMSL represents a source detection algorithm relying on diffusion models, necessitating an assessment of its aptitude for generalization. Concurrently, we assessed its computational complexity while also incorporating experiments utilizing two real diffusion datasets. We have documented the results of these experiments in Appendix G.

**Visualization**. To further showcase the efficacy of DDMSL in solving the inverse problem of information diffusion on complex networks, we designed a visualization that demonstrates source localization and the propagation evolution process. Due to space constraints, we have presented the visualization in Appendix H.

### 4.4 Ablation study

Table 3: The results of the ablation study.

| Methods | Karate | | | | Jazz | | | | Cora Ml | | | | Power Grid | | | | PGP | | | |
|---|---|---|---|---|---|---|---|---|---|---|---|---|---|---|---|---|---|---|---|---|
| | PR | RE | F1 | AUC | PR | RE | F1 | AUC | PR | RE | F1 | AUC | PR | RE | F1 | AUC | PR | RE | F1 | AUC |
| **DDMSL** | 0.708 | 0.736 | 0.722 | 0.853 | 0.817 | 0.881 | 0.848 | 0.930 | 0.894 | 0.867 | 0.880 | 0.928 | 0.833 | 0.879 | 0.855 | 0.930 | 0.856 | 0.903 | 0.879 | 0.943 |
| DDMSL(a) | 0.264 | 0.910 | 0.379 | 0.827 | 0.254 | 0.683 | 0.367 | 0.741 | 0.277 | 0.626 | 0.384 | 0.722 | 0.459 | 0.605 | 0.522 | 0.706 | 0.366 | 0.857 | 0.513 | 0.846 |
| DDMSL(b) | 0.655 | 0.850 | 0.718 | 0.860 | 0.648 | 0.822 | 0.723 | 0.889 | 0.834 | 0.856 | 0.845 | 0.915 | 0.818 | 0.875 | 0.845 | 0.919 | 0.811 | 0.888 | 0.848 | 0.922 |
| DDMSL(c) | 0.339 | 0.185 | 0.236 | 0.592 | 0.931 | 0.111 | 0.1943 | 0.556 | 0.817 | 0.043 | 0.08 | 0.521 | 0.998 | 0.2756 | 0.432 | 0.638 | 0.997 | 0.656 | 0.792 | 0.823 |

To evaluate the effectiveness of each component, we conducted ablation experiments on DDMSL in SIR diffusion mode. One essential component of DDMSL is the reversible residual network. We removed this component and replaced each reversible residual block with a GCN network, denoted as DDMSL(a). Additionally, DDMSL is also supervised by $L_{constarin}$ on the propagation rules. The model without the propagation rule supervision module is denoted as DDMSL(b). Finally, to assess the impact of GCN (Graph Convolutional Network) in reversible residual blocks, DDMSL(c) represents the model after removing the GCN modules from residual blocks. We ran the three variants on five datasets and compared the results,

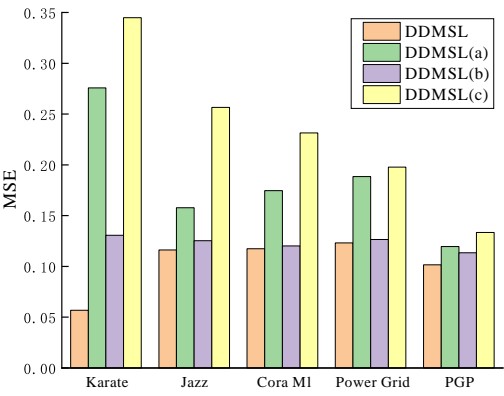

Figure 4: MSE error in ablations.

which are shown in Table 3 and Figure 4. As can be seen in table 3, we have these observations: (1) removing the residual module (DDMSL(a)) leads to a significant degradation in model performance, with the F1 score and AUC decreasing by 49% and 17%, respectively. Although the GCN network is also a form of MPNNs, we noticed that the output of the inference network composed of GCNs were very similar, indicating node feature oversmoothing, and further highlighting the effectiveness of the reversible residual network module; (2) $L_{constraint}$ is also effective, which contributes to 5% and 2% performance improvement in terms of F1 score and AUC, respectively. Similar results can be obtained in Figure 4 regarding the task of reconstructing the information diffusion processes; and (3) upon the drop of the GCNs from the reversible residual network (DDMSL(c)), a noteworthy deterioration model performance was observed, manifesting as a substantial decrease in F1 scores and AUC by 59% and 32% respectively. This compellingly signifies the indispensable role of GCN in acquiring the diffusion mode of the SIR model through the process of learning. Similar results can be obtained in Figure 4 regarding the task of reconstructing the information diffusion processes.

## 5   Conclusions

In this paper, we introduced a reversible residual network block based on the relationship between diffusion phenomena and message passing neural networks, while ensuring the reversibility of the network by limiting its Lipshitz coefficient. Using this, we constructed a discrete denoising diffusion model (DDMSL) which can locate the source of graph diffusion and restore the diffusion paths. Extensive experiments on five real datasets have demonstrated the effectiveness of DDMSL and its constituent modules. Our work offers insights into how to calculate the distribution of solutions to graph diffusion inverse problems based on the information propagation laws on complex networks.

Solving the inverse problem of graph diffusion plays a crucial role in many social operations, including controlling the spread of infectious diseases, rumors, and computer viruses. It provides valuable insights on enhancing source detection performance and fills the gap in methods for recovering diffusion evolution. However, our work has some limitations. For instance, DDMSL requires the prior knowledge about propagation models, such as infection rate and recovery rate. Although we can infer these parameters from existing observation data [32, 21], it limits the application of DDMSL in situations with insufficient observations of propagation conditions. Our future research will focus on reducing the dependence of DDMSL on prior conditions.

## 6   Acknowledgement

This work was supported in part by the National Key R&D Program of China under Grant No. 2021YFC3300302, the National Natural Science Foundation of China (Grant No. 62202089, 72192832 and U22A2004), the General project of Liaoning Provincial Department of Education (Grant No. LJKZ0005), the Shanghai Rising-Star Program (Grant No. 23QA1403100), the Natural Science Foundation of Shanghai (Grant No. 21ZR1421900), and the Fundamental Research Funds for the Central Universities (Grant No. N2319004).

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

## A  Broader impacts

Overall, our work offers valuable insights into how to limit the spread of malicious information. For example, by tracing the spread of infectious diseases, we can identify potential contacts of infected individuals, effectively containing the outbreak. However, it is important to consider the potential negative implications of this approach for society, such as compromising the privacy of individuals living with infectious diseases like HIV. Addressing these concerns should be important.

## B  Derivation of forward and backward processes

**Derivation of forward processes**. Extending Equation 4 by means of the Markov property:

$$
\begin{aligned}
q\left(\boldsymbol{x}_t^i \mid \boldsymbol{x}_0^i\right) &= \sum_{\boldsymbol{x}_{1:t-1}^i} \prod_{k=1}^t q\left(\boldsymbol{x}_k^i \mid \boldsymbol{x}_{k-1}^i\right) \\
&= \sum_{\boldsymbol{x}_{1:t-1}^i} \prod_{k=1}^t \boldsymbol{x}_{k-1}^i Q_k^i \boldsymbol{x}_{\boldsymbol{k}}^{\boldsymbol{i}\,T} \\
&= \sum_{\boldsymbol{x}_{1:t-1}^i} \boldsymbol{x}_0^i Q_1^i \boldsymbol{x}_{\boldsymbol{1}}^{\boldsymbol{i}\,T} \cdots \boldsymbol{x}_{\boldsymbol{k-1}}^{\boldsymbol{i}} Q_k^i \boldsymbol{x}_{\boldsymbol{k}}^{\boldsymbol{i}\,T} \cdots \boldsymbol{x}_{t-1}^i Q_t^i \boldsymbol{x}_{\boldsymbol{t}}^{\boldsymbol{i}\,T} \\
&= \boldsymbol{x}_0^i Q_1^i \left(\sum_{\boldsymbol{x}_1^i} \boldsymbol{x}_{\boldsymbol{1}}^{\boldsymbol{i}\,T} \boldsymbol{x}_1^i\right) \cdots \left(\sum_{\boldsymbol{x}_{t-1}^i} \boldsymbol{x}_{\boldsymbol{t-1}}^{\boldsymbol{i}\,T} \boldsymbol{x}_{t-1}^i\right) Q_t^i \boldsymbol{x}_{\boldsymbol{t}}^{\boldsymbol{i}\,T} \\
&= \boldsymbol{x}_0^i Q_1^i I Q_2^i \cdots I Q_k^i I \cdots I Q_t^i \boldsymbol{x}_{\boldsymbol{t}}^{\boldsymbol{i}\,T} \\
&= \boldsymbol{x}_0^i \bar{Q}_t^i \boldsymbol{x}_{\boldsymbol{t}}^{\boldsymbol{i}\,T} \sim \mathrm{Cat}\left(\boldsymbol{x}_t^i; \boldsymbol{p} = \boldsymbol{x}_0^i \bar{Q}_t^i\right)
\end{aligned}
\tag{18}
$$

Where $I$ is the identity matrix.

**Derivation of backward processes**. The detailed derivation of Equation 6 is as follows:

$$
\begin{aligned}
q\left(\boldsymbol{x}_{t-1}^i \mid \boldsymbol{x}_t^i, \boldsymbol{x}_0^i\right) &= \frac{q\left(\boldsymbol{x}_t^i \mid \boldsymbol{x}_{t-1}^i, \boldsymbol{x}_0^i\right) q\left(\boldsymbol{x}_{t-1}^i \mid \boldsymbol{x}_0^i\right)}{q\left(\boldsymbol{x}_t^i \mid \boldsymbol{x}_0^i\right)} \\
&= \frac{q\left(\boldsymbol{x}_t^i \mid \boldsymbol{x}_{t-1}^i\right) q\left(\boldsymbol{x}_{t-1}^i \mid \boldsymbol{x}_0^i\right)}{q\left(\boldsymbol{x}_t^i \mid \boldsymbol{x}_0^i\right)} \\
&= \frac{\boldsymbol{x}_{t-1}^i Q_t^i \boldsymbol{x}_{\boldsymbol{t}}^{\boldsymbol{i}\,T} \cdot \boldsymbol{x}_0^i \bar{Q}_{t-1}^i \boldsymbol{x}_{\boldsymbol{t-1}}^{\boldsymbol{i}\,T}}{\boldsymbol{x}_0^i \bar{Q}_t^i \boldsymbol{x}_{\boldsymbol{t}}^{\boldsymbol{i}\,T}} \\
&= \frac{\left(\boldsymbol{x}_t^i Q_t^{i\,T} \boldsymbol{x}_{\boldsymbol{t-1}}^{\boldsymbol{i}\,T}\right) \cdot \left(\boldsymbol{x}_0^i \bar{Q}_{t-1}^i \boldsymbol{x}_{\boldsymbol{t-1}}^{\boldsymbol{i}\,T}\right)}{\boldsymbol{x}_0^i \bar{Q}_t^i \boldsymbol{x}_{\boldsymbol{t}}^{\boldsymbol{i}\,T}} \\
&= \frac{\left(\boldsymbol{x}_t^i Q_t^{i\,T}\right) \odot \left(\boldsymbol{x}_0^i \bar{Q}_{t-1}^i\right) \left(\boldsymbol{x}_{\boldsymbol{t-1}}^{\boldsymbol{i}\,T}\right)}{\boldsymbol{x}_0^i \bar{Q}_t^i \boldsymbol{x}_{\boldsymbol{t}}^{\boldsymbol{i}\,T}} \\
&\sim \mathrm{Cat}\left(\boldsymbol{x}_{t-1}^i; \boldsymbol{p} = \frac{\left(\boldsymbol{x}_t^i Q_t^{i\,T}\right) \odot \left(\boldsymbol{x}_0^i \bar{Q}_{t-1}^i\right)}{\boldsymbol{x}_0^i \bar{Q}_t^i \boldsymbol{x}_{\boldsymbol{t}}^{\boldsymbol{i}\,T}}\right)
\end{aligned}
\tag{19}
$$

From the Bayesian formula, it follows that:

$$
\begin{aligned}
q\left(\boldsymbol{x}_{t-1}^i \mid \boldsymbol{x}_t^i, \boldsymbol{x}_0^i\right) = q\left(\boldsymbol{x}_{t-1}^i \mid \boldsymbol{x}_t^i\right) &= \frac{\sum_{\boldsymbol{x}_0^i} q\left(\boldsymbol{x}_{t-1}^i, \boldsymbol{x}_t^i, \boldsymbol{x}_0^i\right)}{q\left(\boldsymbol{x}_t^i\right)} \\
&= \frac{\sum_{\boldsymbol{x}_0^i} q\left(\boldsymbol{x}_{t-1}^i \mid \boldsymbol{x}_t^i, \boldsymbol{x}_0^i\right) q\left(\boldsymbol{x}_0^i \mid \boldsymbol{x}_t^i\right) q\left(\boldsymbol{x}_t^i\right)}{q\left(\boldsymbol{x}_t^i\right)} \\
&= \sum_{\boldsymbol{x}_0^i} q\left(\boldsymbol{x}_{t-1}^i \mid \boldsymbol{x}_t^i, \boldsymbol{x}_0^i\right) q\left(\boldsymbol{x}_0^i \mid \boldsymbol{x}_t^i\right) \\
&= \mathbb{E}_{q\left(\boldsymbol{x}_0^i \mid \boldsymbol{x}_t^i\right)} q\left(\boldsymbol{x}_{t-1}^i \mid \boldsymbol{x}_t^i, \boldsymbol{x}_0^i\right)
\end{aligned}
\tag{20}
$$

Fitting this distribution using a neural network:

$$
\begin{aligned}
q\left(\boldsymbol{x}_{t-1}^i \mid \boldsymbol{x}_t^i\right) \approx p_\theta\left(\boldsymbol{x}_{t-1}^i \mid \boldsymbol{x}_t^i\right) = \mathbb{E}_{\boldsymbol{x}_0^i \sim p_\theta\left(\boldsymbol{x}_0^i \mid \boldsymbol{x}_t^i\right)} &\, q\left(\boldsymbol{x}_{t-1}^i \mid \boldsymbol{x}_t^i, \boldsymbol{x}_0^i\right) \\
&= \mathbb{E}_{p_\vartheta\left(\boldsymbol{x}_0^i \mid \boldsymbol{x}_t^i\right)} \frac{q\left(\boldsymbol{x}_t^i \mid \boldsymbol{x}_{t-1}^i, \boldsymbol{x}_0^i\right) q\left(\boldsymbol{x}_{t-1}^i \mid \boldsymbol{x}_0^i\right)}{q\left(\boldsymbol{x}_t^i \mid \boldsymbol{x}_0^i\right)} \\
&= \mathbb{E}_{p_0\left(\boldsymbol{x}_0^i \mid \boldsymbol{x}_t^i\right)} \frac{q\left(\boldsymbol{x}_t^i \mid \boldsymbol{x}_{t-1}^i\right) q\left(\boldsymbol{x}_{t-1}^i \mid \boldsymbol{x}_0^i\right)}{q\left(\boldsymbol{x}_t^i \mid \boldsymbol{x}_0^i\right)} \\
&= \frac{\sum_j\left[q\left(\boldsymbol{x}_t^i \mid \boldsymbol{x}_{t-1}^i\right) q\left(\boldsymbol{x}_{t-1}^i \mid \boldsymbol{x_0^i}^{(j)}\right) p_\theta\left(\boldsymbol{x_0^i}^{(j)} \mid \boldsymbol{x}_t^i\right)\right]}{q\left(\boldsymbol{x}_t^i \mid \boldsymbol{x}_0^i\right)} \\
&= \frac{q\left(\boldsymbol{x}_t^i \mid \boldsymbol{x}_{t-1}^i\right)\left[\sum_j q\left(\boldsymbol{x}_{t-1}^i \mid \boldsymbol{x_0^i}^{(j)}\right) p_\theta\left(\boldsymbol{x_0^i}^{(j)} \mid \boldsymbol{x}_t^i\right)\right]}{q\left(\boldsymbol{x}_t^i \mid \boldsymbol{x}_0^i\right)}
\end{aligned}
\tag{21}
$$

## C  Propagation rule constraint of information diffusion reconstruction

To investigate the circumstances under which the propagation rules may be violated, let's revisit Equation 6. Note that the denominator serves as the normalization term, while the numerator is composed of two key terms - $\left(\boldsymbol{x}_t^i Q_t^{i\,T}\right)$ and $\left(\boldsymbol{x}_0^i \bar{Q}_{t-1}^i\right)$ - that are crucial in preserving the propagation rule.

The three rows of $Q_t^{i\,T}$ correspond to the distribution of $q(x_t^i \mid x_{t-1}^i, x_0^i)$ when $x_t^i$ is in one of three states: $S$, $I$, and $R$. Similarly, the three rows of $\bar{Q}_t^i$ represent the distribution of $q(x_{t-1}^i \mid x_0^i)$ when $x_0^i$ is in one of three states: $S$, $I$, and $R$. Let $\left[\bar{Q}_t^i\right]_{12}$ and $\left[\bar{Q}_t^i\right]_{13}$ be denoted by $q_t^a$ and $q_t^b$ respectively, then $q_1^a = \gamma_1^i, q_1^b = 0$.

$$
\bar{Q}_t^i = \begin{bmatrix}
\prod_{k=1}^t (1 - \beta_k^i) & \prod_{k=1}^{t-1}(1-\beta_k^i)\beta_t^i + q_{t-1}^a(1-\gamma_t^i) & q_{t-1}^a(\gamma_t^i) + q_{t-1}^b \\
0 & \prod_{k=1}^t(1-\gamma_k^i) & \sum_{j=1}^t \prod_{k=1}^{t-j}(1-\gamma_k^i)\gamma_{k+1}^i \\
0 & 0 & 1
\end{bmatrix}
\tag{22}
$$

When $x_t^i$ and $x_0^i$ are in different states, the results of $q(x_{t-1}^i \mid x_t^i, x_0^i) = \left(\boldsymbol{x}_t^i Q_t^{i\,T}\right) \odot \left(\boldsymbol{x}_0^i \bar{Q}_{t-1}^i\right)$ are shown in Table 4. The table reveals that $q(x_{t-1}^i \mid x_t^i, x_0^i) = [0,0,0]$ when the propagation rule is violated. Since we are predicting $x_0^i$, there are only two possible states for $x_0^i$: $S$ (Susceptible) and $I$ (Infected), with $x_0^i = R$ (Recovered) being excluded. In such instances, $q(x_{t-1}^i \mid x_t^i, x_0^i)$ can be set to $[1,0,0]$ to resolve the issue. Furthermore, if $x_t^i = R$ and $x_0^i = S$, then $x_{t-1}^i = S$ would violate the propagation rules. However, as shown in the $9th$ row of Table 4, the probability of $x_{t-1}^i = S$ is much smaller than that of $x_{t-1}^i = I$ and $x_{t-1}^i = R$, and such situations will not lead to training failure. Hence, there is no need for any specific handling of this scenario.

To further minimize propagation rule violations during the training process, we incorporate supervision of the propagation rule. Specifically, when using this supervision function, nodes that have a state of $R$ are set to $I$ to enforce the propagation rule.

$$
L_{constrain1} = Relu\left(\mathbf{X_{t-1}} - (\mathbf{A} + \mathbf{I})\mathbf{X_0}\right)
\tag{23}
$$

Table 4: The distribution of the unnormalized $q(x_{t-1}^i|x_t^i, x_0^i)$ in different cases.

| $x_t^i$ | $x_0^i$ | $q(x_{t-1}^i|x_t^i, x_0^i)$ S | I | R |
|---|---|---|---|---|
| S | S | $\prod_{k=1}^t(1-\beta_k^i)\prod_{k=1}^{t-1}(1-\beta_k^i)$ | 0 | 0 |
| | I | 0 | 0 | 0 |
| | R | 0 | 0 | 0 |
| I | S | $q_t^a\prod_{k=1}^{t-1}(1-\beta_k^i)$ | $\prod_{k=1}^t(1-\gamma_k^i)q_{t-1}^a$ | 0 |
| | I | 0 | $\prod_{k=1}^t(1-\gamma_k^i)\prod_{k=1}^{t-1}(1-\gamma_k^i)$ | 0 |
| | R | 0 | 0 | 0 |
| R | S | $q_t^b\prod_{k=1}^{t-1}(1-\beta_k^i)$ | $q_{t-1}^a\sum_{j=1}^t\prod_{k=1}^{t-j}(1-\gamma_k^i)\gamma_{k+1}^i$ | $q_{t-1}^b$ |
| | I | 0 | $\sum_{j=1}^t\prod_{k=1}^{t-j}(1-\gamma_k^i)\gamma_{k+1}^i \cdot \prod_{k=1}^{t-1}(1-\gamma_k^i)$ | $\sum_{j=1}^t\prod_{k=1}^{t-j}(1-\gamma_k^i)\gamma_{k+1}^i$ |
| | R | 0 | 0 | 1 |

where $(\mathbf{A}+\mathbf{I})\mathbf{X_0}$ represents the total number of infected nodes in a given node's first-order neighborhood. Equation 23 penalizes $\mathbf{X_0}$ if the node is infected and there are no other infected nodes within its first-order neighborhood. To maintain the stability of inferred $\mathbf{X_0}$ generating $\mathbf{X_{t-1}}$, we apply the same constraint to the process. Specifically, we utilize the monotonicity regularization of information diffusion from [22]. If the source set $\mathbf{X_0^{(i)}}$ is a superset of $\mathbf{X_0^{(j)}}$, then the generated $\mathbf{X_{t-1}}$ resulting from their diffusion needs to satisfy the following equation.

$$L_{constrain2} = \left\|\max\left(\mathbf{0}, \mathbf{X_{t-1}^{(j)}} - \mathbf{X_{t-1}^{(i)}}\right)\right\|^2, \forall \mathbf{X_0^{(i)}} \supseteq \mathbf{X_0^{(j)}} \tag{24}$$

# D  Proofs of lemmas and theorems

## D.1  The proof of theorem 3.1

**Proof** *Set the initial infection seed set as:* $\mathbb{S} = \{x_0^{s_1}, x_0^{s_2}, \ldots x_0^{s_m}\}$. *At the initial moment, the infection status distribution of node* $i$ *is:*

$$\begin{cases} P_S^{i=S_m}(0) = 0, P_I^{i=S_m}(0) = 1 \\ P_S^{i \neq S_m}(0) = 1, P_I^{i \neq S_m}(0) = 0 \\ P_R^i(0) = 0 \end{cases} \tag{25}$$

*At time* $t$, *the infection status distribution of node* $i$ *is:*

$$\begin{cases} P_I^i(t) = P_I^i(t-1)(1-\gamma_R^i(t-1)) + P_S^i(t-1)\left[1 - \prod_j(1-\beta_I^j(t))A_{ij}P_I^j(t-1)\right] \\ P_S^i(t) = P_S^i(t-1)\left[\prod_j(1-\beta_I^j(t))A_{ij}P_I^j(t-1)\right] \\ P_R^i(t) = P_I^i(t-1)\gamma_R^i(t) \end{cases} \tag{26}$$

*where A is the adjacency matrix and* $j$ *is the neighbor of node* $i$. *When dealing with a static graph* $\mathcal{G}$, *A is fixed, allowing for determination of the state distribution at any time based on the initial node state. Specifically, if both graph G and the seed node set* $\mathbb{S}$ *are known, it becomes possible to calculate the state distribution of each node at any given time using Equation 26. Utilizing Equations 2 and 3,* $Q_t^i$ *can also be determined.*

## D.2  Proof of lemma 3.1

**Proof** *The graph convolution layer with batch normalization* $\mathbf{BN}(g(\xi))$ *can be abbreviated as* $GConv$. *In our approach, we apply spectral normalization to both the linear transformation* $\mathbf{U}$ *and convolutional layers* $GConv$. *As a result, the weight parameters of both the networks* $f_w$ *and* $GConv$ *possess 1-Lipschitz continuity after spectral normalization, as described in [28].*

Let $GConv : \mathbb{R}^n \to \mathbb{R}^m$, $x_1, x_2 \in \mathbb{R}^n$. Note that the nonlinear activation function $\sigma(\cdot)$ is set to $Mish(\cdot)$, which obviously possesses 1-Lipschitz continuity.

$$
\begin{aligned}
\|f_\theta(x_1) - f_\theta(x_2)\|_p &= \|\sigma(GConv(x_1)) - \sigma(GConv(x_2))\|_p \\
&\leq \|GConv(x_1) - GConv(x_2)\|_p \\
&< \|x_1 - x_2\|_p
\end{aligned}
\tag{27}
$$

where $\|\cdot\|_p$ represents the p-norm ($p = 2$ or $p = \infty$). Therefore, the Lipschitz constant of $f_\theta$ is less than 1.

### D.3 Proof of theorem 3.2

**Proof** *To prove that $\mathbf{Y_T} = \mathbf{F}(f_w(\mathbf{X_0}))$ is reversible, it is necessary to ensure that $F$ and $f_w$ are reversible [43].*

$$
\begin{cases}
f_w(\mathbf{X_0}) = \mathbf{X_0} + f_w(\mathbf{X_0}) - \mathbf{X_0} = \xi \\
\xi + f_\theta(\xi) = \mathbf{Y_T}
\end{cases}
\Leftrightarrow
\begin{cases}
\mathbf{X_0} = \xi + \mathbf{X_0} - f_w(\mathbf{X_0}) \\
\xi = \mathbf{Y_T} - f_\theta(\xi)
\end{cases}
\tag{28}
$$

*We construct the following iterative formula:*

$$
\begin{cases}
\mathbf{X_0^{k+1}} = \xi + \mathbf{X_0^k} - f_w(\mathbf{X_0^k}), \mathbf{X_0^0} = \xi \\
\xi^{k+1} = \mathbf{Y_T} - f_\theta(\xi^k), \xi^0 = \mathbf{Y_T}
\end{cases}
\Rightarrow
\begin{cases}
\lim_{k\to\infty} \mathbf{X_0^k} = \mathbf{X_0} \\
\lim_{k\to\infty} \xi^k = \xi
\end{cases}
\tag{29}
$$

*By Lemma 2, we can ensure that $L_w < 1$ and $L_\theta < 1$. Moreover, the Lipschitz constant of $(\mathbf{X_0^k} - f_w(\mathbf{X_0^k})$ is $1 - L_w$, which is less than 1. Thus, when the number of iterations $k$ is sufficiently large, it follows that the transformation $\mathbf{Y_T} = \mathbf{F}(f_w(\mathbf{X_0}))$ is reversible according to the Banach fixed point theorem [4].*

### D.4 Proof of theorem 3.3

**Proof** *Specifically, Theorem 3.2 proves that $(\mathbf{F_1} \circ \mathbf{F_2} \circ \ldots \circ \mathbf{F_n})(\xi)$ is invertible when $n = 1$. Therefore, we are currently examining whether $(\mathbf{F_1} \circ \mathbf{F_2} \circ \ldots \circ \mathbf{F_n})(\xi)$ retains its reversibility for $n > 1$. To make the notation simpler, we denote $(\mathbf{F_1} \circ \mathbf{F_2} \circ \ldots \circ \mathbf{F_i})(\xi)$ as $\hat{\mathbf{F_i}}$.*

$$
\hat{\mathbf{F_n}} = (\mathbf{F_1} \circ \mathbf{F_2} \circ \ldots \circ \mathbf{F_n})(\xi) = \underbrace{\mathbf{DP} \circ \cdots \circ \mathbf{DP}}_{n} \left[ \xi + f_w(\xi) + \sum_{i=1}^{n-1} f_w\left(\hat{\mathbf{F_i}}(\xi)\right) \right]
\tag{30}
$$

*The application of the dropout function $\mathbf{DP}$ will limit the Lipschitz constant of any function $f$ to $1 - r$ times its original value: $L_{\mathbf{DP}(f)} = (1-r)L_f$. Additionally, we have $L_{\hat{\mathbf{F_i}}} \leq \prod_{j=1}^{i} \left(L_{F_j}\right) \leq (1-r)^i(1 + L_{f_w})^i$ [13]. Therefore, the Lipschitz constant of $\hat{\mathbf{F_n}}$ is expressed as:*

$$
\begin{aligned}
L_{\hat{\mathbf{F_n}}} &\leq (1-r)^n \left[ 1 + L_{f_w} + \sum_{i=1}^{n-1} L_{f_w} \cdot L_{\hat{\mathbf{F_i}}} \right] \\
&\leq (1-r)^n \left[ 1 + L_{f_w} \sum_{i=0}^{n-1} [(1-r)(1+L_{f_w})]^i \right] \\
&= (1-r)^n \left[ 1 + L_{f_w} \cdot \frac{1 - [(1-r)(1+L_{f_w})]^n}{1 - (1-r)(1+L_{f_w})} \right]
\end{aligned}
\tag{31}
$$

*Note that when $L_{f_w} < 1$ and $n > 1$, we only need to set $r = 1/2$ to ensure that $L_{\hat{\mathbf{F_n}}} < 1$, hereby guaranteeing that $\hat{\mathbf{F_n}}$ is reversible.*

## E   DDMSL implementation details

The DDMSL approach has been previously explained, and we will now provide further details on the implementation of DDMSL. The linear transform $\mathbf{U}$ refers to a fully connected layer, and in Figure 2, the dense layer is composed of two fully connected layers that undergo spectral normalization. The

final output of the $nn_\theta$ is represented by an $N \times 1$ matrix, indicating the probability that each node is in an infected state at $t = 0$. We designate nodes with an infection probability higher than a certain threshold as infected nodes.

Given a complete information diffusion instance $\mathbf{X} = \{\mathbf{X_0}, \ldots, \mathbf{X_T}\}$ where $\mathbf{X_t} = \{x_t^1, \ldots, x_t^N\}$, we sample $t \in \{1, \ldots, T\}$ to be included in the neural network $nn_\theta$ based on the probability distribution $p(t) \sim \frac{t}{\sum_{t=1}^T t}$, and use the sine-cosine position encoding [41] to embed $t$. The training and inference processes are shown in Algorithm 1 and Algorithm 2, respectively. Additionally, the variable $T$ remains consistent with the information diffusion step size.

---

**Algorithm 1:** Training

**Input:** $\mathbf{X_0}, \mathcal{G}, \mathbf{X_t}$, threshold $\alpha$
1 **repeat**
2 $\quad \mathbf{Q_t} \leftarrow$ Equation 2 and Equation 3
3 $\quad q(x_{t-1}|x_t) \leftarrow$ Calculate Equation 6 using $\mathbf{X_t}$ and $\mathbf{Q_t}$
4 $\quad t \sim \mathrm{P}(\{1, \ldots, T\}), P(t) = \frac{t}{\sum_{t=1}^T t}$
5 $\quad \mathbf{X_0^{t-1}} = nn_\theta(\mathcal{G}, \mathbf{X_t}, t)$
$\quad$ // Using data from $t$, $nn_\theta$ infers the source node $\mathbf{X_0^{t-1}}$, which is then
$\qquad$ used to reconstruct the diffusion graph at $t-1$.
6 $\quad \mathbf{X_0^{t-1}}[\mathbf{X_0^{t-1}} > \alpha] = 1$
7 $\quad \mathbf{X_0^{t-1}}[\mathbf{X_0^{t-1}}! = 1] = 0$
8 $\quad \mathbf{Q_t^{'}} \leftarrow$ Equation 2 and Equation 3
$\quad$ // $\mathbf{Q_t^{'}}$ is generated by $\mathbf{X_0^{t-1}}$.
9 $\quad P_\theta(x_{t-1}|x_t) \leftarrow$ Calculate Equation 8 using $\mathbf{X_t}, \mathbf{X_0^{t-1}}$ and $\mathbf{Q_t^{'}}$
10 $\quad \mathbf{X_{t-1}} \leftarrow Gumbel - Softmax(P_\theta(x_{t-1}|x_t))$
11 Take gradient descent step on:
12 $\nabla_\theta(L_{simple} + L_{constrain})$

---

**Algorithm 2:** Infering

**Input:** $\mathbf{X_t}, \mathcal{G}$, Empty set $\mathbb{X}$, threshold $\alpha$
**Output:** $\mathbb{X}$
1 **for** $t$ *starts from* $T$ *to* $1$ **do**
2 $\quad \mathbf{X_0^{t-1}} = nn_\theta(\mathcal{G}, \mathbf{X_t}, t)$
3 $\quad \mathbf{X_0^{t-1}}[\mathbf{X_0^{t-1}} > \alpha] = 1$
4 $\quad \mathbf{X_0^{t-1}}[\mathbf{X_0^{t-1}}! = 1] = 0$
5 $\quad \mathbf{Q_t^{'}} \leftarrow$ Equation 2 and Equation 3
6 $\quad P_\theta(x_{t-1}|x_t) \leftarrow$ Calculate Equation 8 using $\mathbf{X_t}, \mathbf{X_0^{t-1}}$ and $\mathbf{Q_t^{'}}$
7 $\quad \mathbf{X_{t-1}} \leftarrow Gumbel - Softmax(P_\theta(x_{t-1}|x_t))$
8 $\quad \mathbb{X}[t] \leftarrow \mathbf{X_{t-1}}$
9 **end**

---

In Algorithm 1, we have $\mathbf{Q_t} = \{Q_t^1, \ldots, Q_t^N\}$, where $Q_t^i$ can be calculated using Equations 2 and 3. Additionally, the $P_S^i(t)$, $P_I^i(t)$, and $P_R^i(t)$ in Equation 3 can be computed using various methods. Monte Carlo simulations provide the most accurate results, but require at least $10^5$ simulations to be sufficiently precise, leading to a high time complexity. An alternative approach is to utilize a neural network model [10, 47] to learn $\left[P_S^i(t), P_I^i(t), P_R^i(t)\right]$, which significantly reduces the training time of the model. When applied to the SI model, DDMSL utilizes the state transfer matrix by:

$$Q_t^i = \begin{bmatrix} 1 - \beta_I^i(t) & \beta_I^i(t) \\ 0 & 1 \end{bmatrix} \tag{32}$$

where $\mathbf{X_0^{t-1}}$ represents the predicted source node using $\mathbf{X_t}$, while $Q_t^{'}$ is generated using the same method as above. Ultimately, $P_\theta(x_{t-1}|x_t)$ is calculated using $\mathbf{X_0^{t-1}}, \mathbf{X_t}$, and $\mathbf{X_0^{t-1}}$. We obtain $\mathbf{X_{t-1}}$ by sampling from $P_\theta(x_{t-1}|x_t)$, and label nodes as $0(S)$, $1(I)$, or $2(R)$ based on their state. Algorithm 2 proceeds in a similar manner to the process described above.

## F Additional algorithms and dataset parameters

### F.1 Hyperparameters of the algorithms

The hyperparameters for each algorithm have been set according to the values shown in Table 5. Any parameter that is not stated as default is common to both SI and SIR models. In the updated version of SLVAE, the original three-layer MLP network encoder was replaced with a three-layer GCN network, resulting in improved performance. For hyperparameters and implementation details of other algorithms, please refer to the corresponding original papers. DDMSL and deep learning comparison

Table 5: Hyperparameter settings of different algorithms.

| Algorithms | Hyper-parameter | karate | jazz | cora_ml | power grid | PGP | Search space | Description |
|---|---|---|---|---|---|---|---|---|
| DDMSL | Initial learning rate | $2 \times 10^{-3}$ | $2 \times 10^{-3}$ | $2 \times 10^{-3}$ | $2 \times 10^{-3}$ | $3 \times 10^{-3}$ | $[2 \times 10^{-3}, 4 \times 10^{-3}, 5 \times 10^{-3}]$ | Learning rate decreases by 0.97 times the set epoach. |
| | Learning Rate Decline Interval | [1200,1500] | [200,1000] | [500,1200] | [500,1200] | [200,500,800,1200] | Determined by the $LOSS$ curves of the training and validation sets. | |
| | n | 6 | 6 | 6 | 6 | 8 | $[min=3, max=9, step=1]$ | Number of residual blocks |
| | Dropout rate | 0.5 | 0.5 | 0.5 | 0.5 | 0.5 | Determined by Theorem \3.3 | |
| | $\alpha$ in SIR model | 0.4 | 0.6 | 0.4 | 0.4 | 0.45 | [min=0.3,max=0.7,step=0.05] | Threshold |
| | $\alpha$ in SI model | 0.4 | 0.45 | 0.4 | 0.4 | 0.45 | [min=0.3,max=0.7,step=0.05] | |
| | Epoch | 2000 | 1600 | 1600 | 1600 | 1600 | Determined by the $LOSS$ curves of the training and validation sets. | |
| SLVAE | $\alpha$ | 0.55 | 0.5 | 0.55 | 0.55 | 0.45 | [min=0.3,max=0.7,step=0.05] | Threshold |
| | Learning rate | $2 \times 10^{-3}$ | $2 \times 10^{-3}$ | $2 \times 10^{-3}$ | $2 \times 10^{-3}$ | $3 \times 10^{-3}$ | $[2 \times 10^{-3}, 4 \times 10^{-3}, 5 \times 10^{-3}]$ | |
| | GCN-based encoder parameters | [64,128,256] | [64,128,256] | [64,128,256] | [64,128,256] | [64,128,256] | [64,128,256],[128,256,512] | The hidden dim of the encoder |
| | MLP-based decoder parameters | [256,128,1] | [256,128,1] | [256,128,1] | [256,128,1] | [256,128,1] | [256,128,1],[512,256,1] | The hidden dim of the decoder |
| | Epoch | 200 | 200 | 200 | 200 | 200 | \ | |
| DDMIX | $\alpha$ | 0.5 | 0.5 | 0.5 | 0.5 | 0.5 | $[min=3, max=9, step=1]$ | Threshold |
| | Learning rate | $2 \times 10^{-3}$ | $2 \times 10^{-3}$ | $2 \times 10^{-3}$ | $2 \times 10^{-3}$ | $2 \times 10^{-3}$ | $[2 \times 10^{-3}, 4 \times 10^{-3}, 5 \times 10^{-3}]$ | |
| | Epoch | 100 | 100 | 100 | 100 | 100 | \ | |

algorithms are both running on Windows 10 systems and trained using a 4090 graphics card. The code for DDMSL is already open source, please refer to: `https://github.com/Ashenone2/DDMSL`.

### F.2 Additional details of the datasets

The description of the data sets used for the experiments and their statistics are shown as below:

- **Karate** [48]: It includes a network of interrelationships between 34 members of the Karate club, comprising 34 nodes and 78 edges. The Karate dataset is a real dataset, widely employed in complex network community discovery research.
- **Jazz** [12]: The Jazz dataset is a network dataset that captures the collaborative relationships between jazz musicians. It comprises 198 nodes and 2,742 edges, and has been extensively used in research on complex network community discovery, node importance metrics, and other related studies.
- **Cora-ML** [27]: Cora-ML is a citation network dataset containing papers from the field of machine learning. Nodes represent papers and edges represent citation relationships between papers.
- **Power Grid** [45]: The Power Grid dataset is a network dataset describing the topology of the Northeastern US power grid, containing 4,941 nodes and 6,594 edges.
- **PGP** [5]: It is a User network of the Pretty-Good-Privacy algorithm for secure information exchange, consisting of 10,680 nodes and 24,316 edges.

Table 6: Statistics of the five datasets.

| Datasets | #Nodes | #Edges | #Avg(degree) | #Average clustering coefficient | #Density | #Diameter |
|---|---|---|---|---|---|---|
| Karate | 34 | 78 | 2.29 | 0.57 | 0.14 | 5 |
| Jazz | 198 | 2,742 | 27.70 | 0.62 | 0.14 | 6 |
| Cora_ml | 2,810 | 7,981 | 5.68 | 0.28 | 0.002 | 17 |
| Power Grid | 4,941 | 6,594 | 1.33 | 0.08 | 0.005 | 46 |
| PGP | 10,680 | 24,316 | 4.55 | 0.27 | 0.0004 | 24 |

# G   Additional experiments

**Experiments on Real Diffusion Datasets**.  In order to gauge the efficacy of DDMSL on real-world propagation datasets, we opted for the Twitter [6] and Douban [6] datasets, encompassing 12,627 nodes with 309,631 edges, and 23,123 nodes with 348,280 edges, respectively. The detailed performance metrics can be found in Table 7.

Table 7: Additional experiments on real diffusion datasets.

| Methods | Twitter | | | | Douban | | | |
|---|---|---|---|---|---|---|---|---|
| | PR | RE | F1 | AUC | PR | RE | F1 | AUC |
| **DDMSL** | 0.445 | 0.286 | 0.313 | 0.625 | 0.484 | 0.324 | 0.381 | 0.622 |
| SLVAE | 0.310 | 0.317 | 0.253 | 0.578 | 0.412 | 0.140 | 0.209 | 0.547 |

**Generalization Performance**. We conducted extensive tests on datasets of varying scales to assess the generalization performance of both DDMSL and SLVAE algorithms. The comparative results are presented in Table 8, revealing that DDMSL demonstrates commendable generalization performance across a majority of scenarios.

Table 8: Additional generalization experiments: Test results on different network topologies after one training on a real network, where the original performance represents the test performance of DDMSL on real networks.

| Training data | Network | Cora Ml | | | | Power Grid | | | | PGP | | | | Twitter | | | | Douban | | | |
|---|---|---|---|---|---|---|---|---|---|---|---|---|---|---|---|---|---|---|---|---|---|
| | | PR | RE | F1 | AUC | PR | RE | F1 | AUC | PR | RE | F1 | AUC | PR | RE | F1 | AUC | PR | RE | F1 | AUC |
| Original performance | DDMSL | 0.790 | 0.908 | 0.845 | 0.941 | 0.763 | 0.966 | 0.852 | 0.966 | 0.754 | 0.887 | 0.815 | 0.928 | 0.445 | 0.286 | 0.313 | 0.625 | 0.484 | 0.324 | 0.381 | 0.622 |
| | SLVAE | 0.721 | 0.765 | 0.852 | 0.908 | 0.908 | 0.719 | 0.803 | 0.856 | 0.817 | 0.721 | 0.766 | 0.851 | 0.310 | 0.317 | 0.253 | 0.578 | 0.412 | 0.140 | 0.209 | 0.547 |
| Small World | DDMSL | 0.732 | 0.826 | 0.776 | 0.896 | 0.812 | 0.499 | 0.62 | 0.744 | 0.987 | 0.684 | 0.808 | 0.841 | 0.375 | 0.299 | 0.290 | 0.618 | 0.409 | 0.295 | 0.301 | 0.624 |
| | SLVAE | 0.824 | 0.576 | 0.678 | 0.781 | 0.626 | 0.335 | 0.436 | 0.656 | 0.982 | 0.539 | 0.696 | 0.769 | 0.308 | 0.241 | 0.227 | 0.572 | 0.375 | 0.124 | 0.186 | 0.544 |
| ER | DDMSL | 0.722 | 0.584 | 0.645 | 0.779 | 0.349 | 0.687 | 0.463 | 0.773 | 0.997 | 0.614 | 0.632 | 0.731 | 0.439 | 0.289 | 0.309 | 0.624 | 0.320 | 0.367 | 0.307 | 0.614 |
| | SLVAE | 0.894 | 0.586 | 0.708 | 0.789 | 0.747 | 0.409 | 0.528 | 0.697 | 0.956 | 0.555 | 0.703 | 0.776 | 0.310 | 0.311 | 0.285 | 0.569 | 0.368 | 0.118 | 0.179 | 0.537 |
| BA Tree | DDMSL | 0.482 | 0.832 | 0.609 | 0.866 | 0.872 | 0.774 | 0.82 | 0.881 | 0.947 | 0.672 | 0.786 | 0.834 | 0.327 | 0.377 | 0.323 | 0.612 | 0.451 | 0.289 | 0.314 | 0.623 |
| | SLVAE | 0.961 | 0.577 | 0.721 | 0.787 | 0.939 | 0.399 | 0.560 | 0.698 | 0.993 | 0.548 | 0.708 | 0.774 | 0.252 | 0.285 | 0.193 | 0.517 | 0.312 | 0.126 | 0.180 | 0.525 |
| BA Dense | DDMSL | 0.749 | 0.683 | 0.715 | 0.829 | 0.654 | 0.354 | 0.459 | 0.667 | 0.972 | 0.598 | 0.741 | 0.798 | 0.369 | 0.305 | 0.288 | 0.622 | 0.423 | 0.289 | 0.304 | 0.623 |
| | SLVAE | 0.662 | 0.611 | 0.635 | 0.788 | 0.731 | 0.441 | 0.550 | 0.712 | 0.997 | 0.446 | 0.617 | 0.723 | 0.286 | 0.405 | 0.315 | 0.567 | 0.374 | 0.132 | 0.195 | 0.553 |

**Time Complexity**. Lastly, we conducted a comparative evaluation of the time complexity between DDMSL and baseline algorithms across diverse datasets, revealing the outcomes illustrated in Table 9. Owing to the gradual inference of diffusion state for each time step, the time complexity of DDMSL tends to be substantial. However, optimization through parallel computing can effectively mitigate this disparity.

Table 9: Additional Time complexity experiment.

| Test time | Cora-Ml | Power-Grid | PGP |
|---|---|---|---|
| **DDMSL** | 15.84s | 22.19s | 22.72s |
| SLVAE | 4.77s | 7.28s | 11.29s |
| DDMIX | 9.34s | 15.7s | 23.26s |
| GCNSI | 1.4s | 8.14s | 15.41s |
| LPSI | 2m14s | 1m51s | 21m37s |
| OJC | 6m11s | 50m17s | 2h41m52s |
| NetSleuth | 2m40s | 4m39s | 21m24s |
| **Training Time** | **Cora-Ml** | **Power-Grid** | **PGP** |
| **DDMSL** | 10m36s | 16m57s | 32m03s |
| SLVAE | 18s | 39s | 1m10s |
| DDMIX | 16m5s | 24m17s | 37m53s |
| GCNSI | 3m53 | 20m6s | 34m26s |

# H Visualization

## H.1 Visualization of reconstructing diffusion paths

To conserve space, we displayed the actual node states and corresponding prediction results every 20% of the time. Figures 5 to 9 showcase the results, where the blue nodes denote susceptible ones, the red nodes denote infected nodes, and the green nodes denote recovered nodes. The findings indicate that DDMSL accurately restored the node states at different times. In contrast, DDMIX could only restore infected nodes, revealing that DDMSL far surpasses DDMIX in its expression capability.

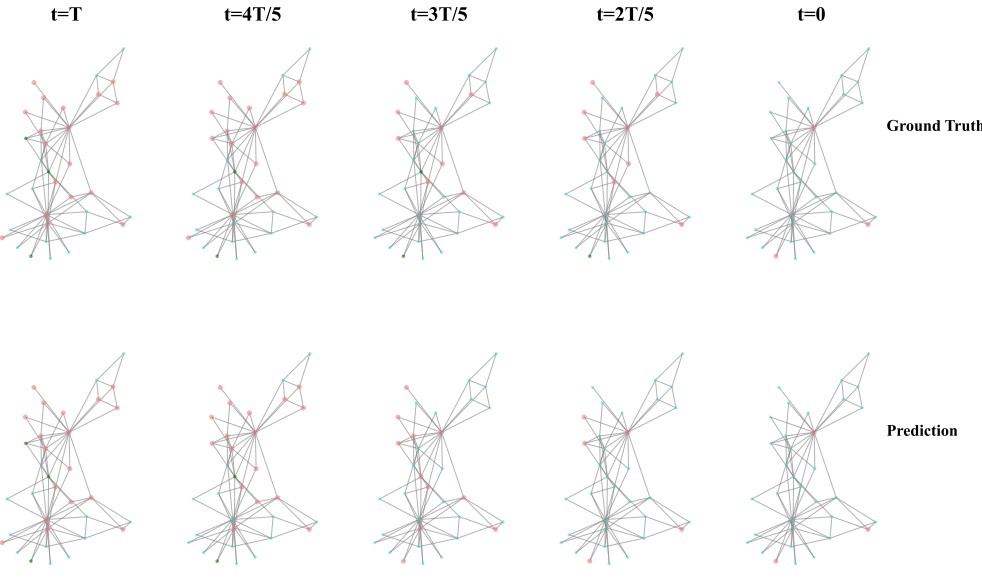

Figure 5: DDMSL reconstructs SIR diffusion on Karate.

## H.2 Visual comparison of source localization

Due to spatial limitations, we only presented the source localization results of DDMSL and benchmark algorithms on the Karate and Jazz datasets. The results are depicted in Figures 10 and 11. The baseline algorithm's performance is unsatisfactory, as evidenced by significant positioning errors in the source nodes. On the contrary, DDMSL outperforms other benchmark algorithms in accurately identifying source nodes. Moreover, even when misidentifying source nodes, DDMSL locates them near the actual nodes.

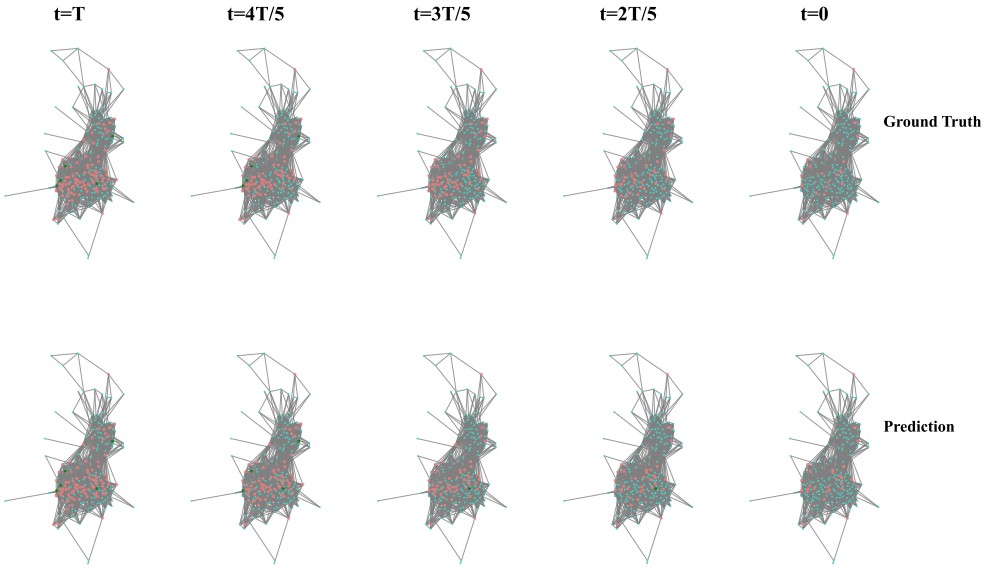

Figure 6: DDMSL reconstructs SIR diffusion on Jazz.

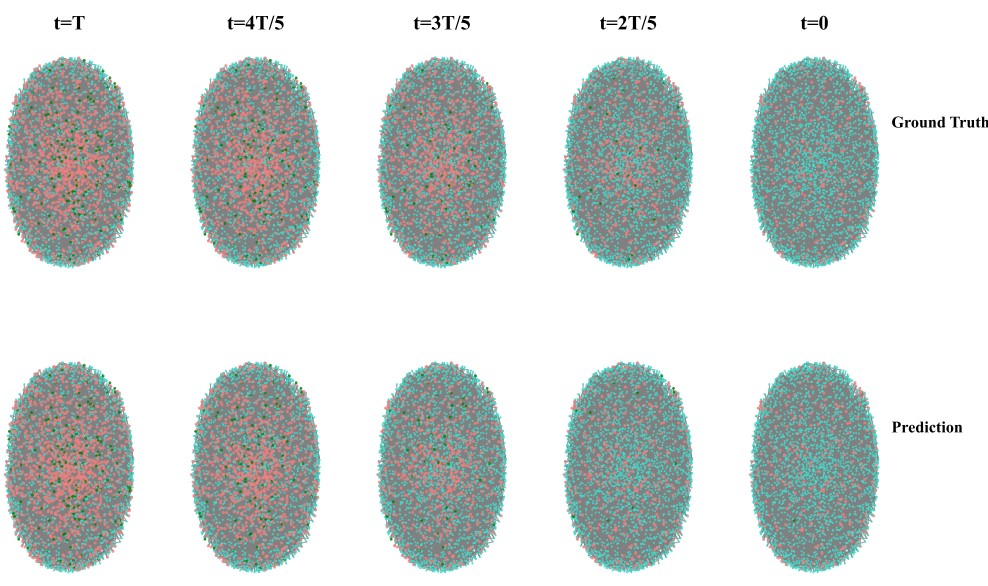

Figure 7: DDMSL reconstructs SIR diffusion on Coral ml.

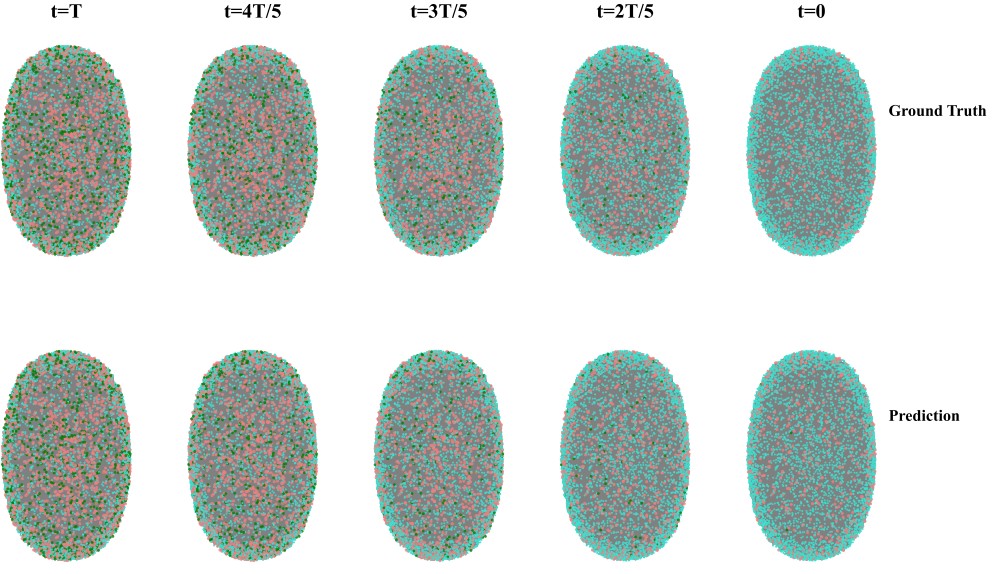

Figure 8: DDMSL reconstructs SIR diffusion on Power grid.

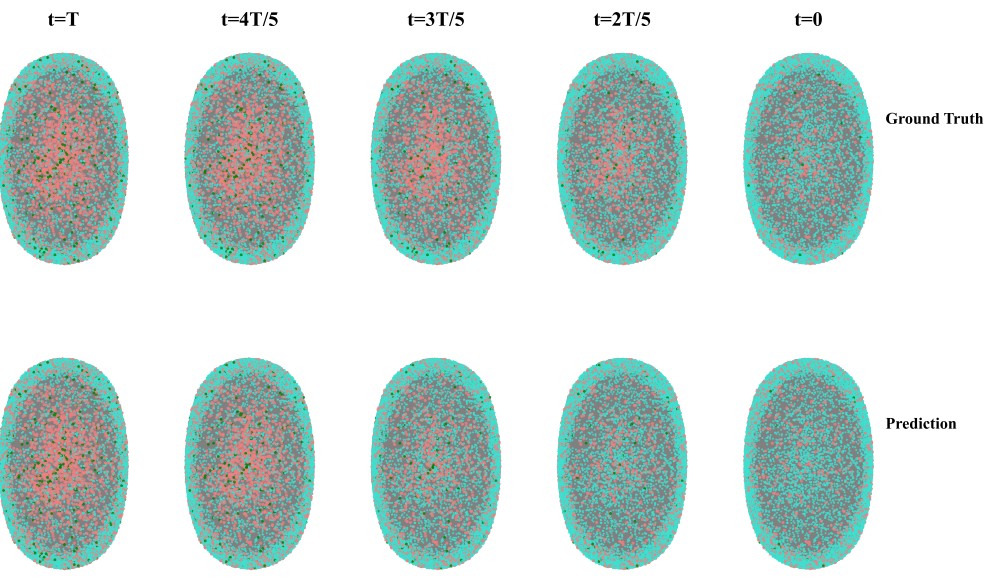

Figure 9: DDMSL reconstructs SIR diffusion on PGP.

**(a) DDMSL**  **(b) SLVAE**  **(c) DDmix**  **(d) LPSI**

**(e) GCNSI**  **(f) OJC**  **(g) Ground Truth**

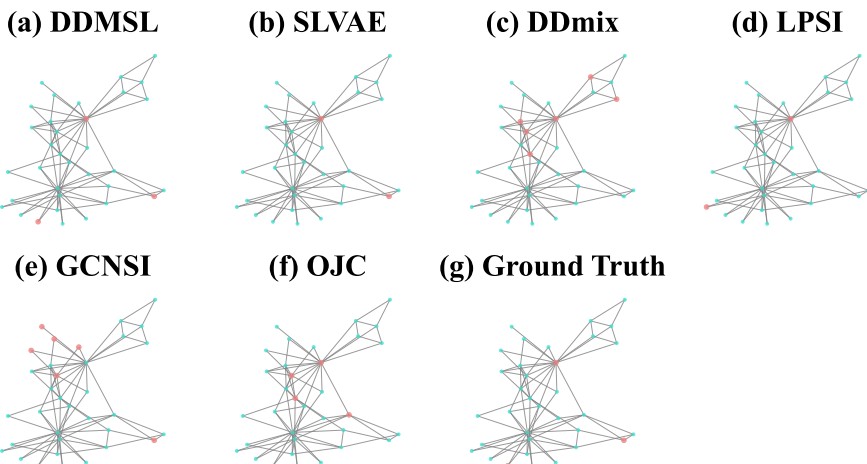

Figure 10: Visualization comparisons of source localization on Karate.

**(a) DDMSL**  **(b) SLVAE**  **(c) DDmix**  **(d) LPSI**

**(e) GCNSI**  **(f) OJC**  **(g) Ground Truth**

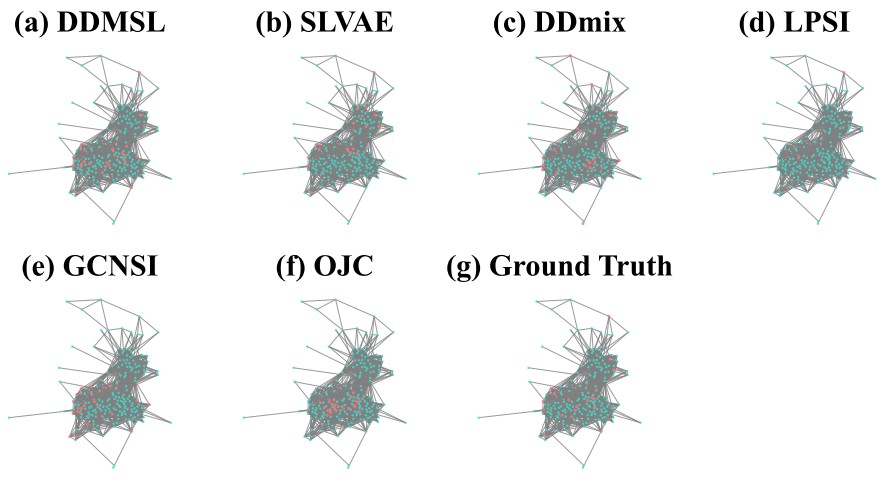

Figure 11: Visualization comparisons of source localization on Jazz.

