# OpenReview forum: "Diffusion Model for Graph Inverse Problems: Towards Effective Source Localization on Complex Networks"
_NeurIPS.cc/2023/Conference — NeurIPS 2023 poster_

### Official Review · Reviewer_c2eK · 2023-07-01

**Soundness:** 3 good
**Presentation:** 3 good
**Contribution:** 3 good
**Rating:** 6
**Confidence:** 4

**Summary:**

This work discusses the challenges associated with tracing the origin and path of information diffusion in complex networks, such as those involved in epidemics or rumors. To address these, the authors propose a probabilistic model, DDMSL (Discrete Diffusion Model for Source Localization), which utilizes Markov chains to model information diffusion and a reversible residual network to localize the source and reconstruct the paths of information diffusion, with its efficacy supported by extensive tests on five real-world datasets.

**Strengths:**

The general topic is interesting, and the paper studies the problem of source localization under two information propagation patterns: SI and SIR. The proposed methods are sound with theoretical guarantee. The overall writing is easy to follow, but the use of notations are sometimes less formal. The idea of simultaneously handling the problems of source localization and reconstructing diffusion paths seems to be interesting, but the motivation is less promising. The experiments are strong, and the proposed method is better than most of existing baselines.

**Weaknesses:**

There are three main questions of this work. Please refer to the questions section.

**Questions:**

1. The motivation for recovering the diffusion path is weak. I feel like the source localization only needs to predict the correct set of seed nodes in order to identify the culprit, why would it be necessary to recover the diffusion paths with the already given diffusion status/observation? I am still having a hard to convince myself with the provided motivation.
2. The authors claim "existing inference results are often deterministic and fail to quantify the uncertainty of source localization", but it seems like the proposed method's results are also deterministic from Figure 2. Does the proposed method output a range of possible source nodes with [S, I, R] labels?
3. In terms of the experiment section, the biggest problem is the efficiency issue of the proposed framework. With the step-wise diffusion model to recover the overall diffusion path, the runtime of the proposed method and the complexity analysis should be demonstrated.

**Limitations:**

I am not seeing any potential negative societal impact.

---

> ### Author Rebuttal · Authors · 2023-08-08
>
>
> Thank you for your valuable suggestion. We have incorporated the time complexity analysis of the DDMSL algorithm and will provide comprehensive answers to all your inquiries.
>
> (1)"The motivation for recovering the diffusion path"
>
> Thank you for your question. In our perspective, reconstructing communication paths plays a pivotal role in controlling information dissemination. This process involves speculating on the distribution of node states throughout the diffusion process.
>
> First, the reconstruction of the diffusion path offers valuable support in managing information spread. When provided with the observed infection status graph $X_T$, early source identification is crucial when T is small as the information diffusion scale is limited, allowing effective control through timely identification. However, as T grows larger, information has already propagated to various parts of the network, making it challenging to effectively control its dissemination even with source identification. In such cases, focus shifts to previously infected nodes or those with a high infection probability.  For instance, in the context of the global Covid-19 pandemic, a common control strategy following a major outbreak involves identifying infected individuals and their close contacts at $T-1$, $T-2$, and $T-s$ based on observed infections in $X_T$, and subsequently isolating them. This strategy has proven to be highly effective in controlling epidemic outbreaks, despite being time-consuming and labor-intensive. Here, our reconstruction of node states at $T-1$, $T-2$, ... and $T-s$ can provide direct assistance in this context.
>
> Additionally, reconstructing the diffusion path can deepen our understanding of information diffusion processes in complex networks and offer insights applicable to other fields. DDMSL generates possible diffusion paths from observed graphs, guiding network control for specific information. It has applications in disease and rumor control, as well as influence maximization. The research on reconstructing diffusion pathways holds immense practical value, and we are optimistic about its future prospects.
>
> (2)"Does DDMSL output a range of possible source nodes with [S,I,R] labels?"
>
> The answer to this question is affirmative.DDMSL can generate a series of possible source nodes, each accompanied by probability labels representing three states:S,I, and R.
>
> In practice, we can only observe $X_t$, which corresponds to the infection data.Therefore,we must infer $X_{t-1}$ from $X_t$ iteratively until we obtain the inferred result of $X_0$.
>
> Various methods exist for inferring $X_{t-1}$ from $X_t$. For example, DDIM predicts the noise between $X_t$ and $X_{t-1}$ and subtracts it from $X_t$ to obtain $X_{t-1}$. In our case, we use a discrete denoising diffusion model (D3PM), which utilizes a neural network model to predict an intermediate $X_0^{'}$ (different from $X_0$) based on $X_t$. Then, $X_0^{'}$ is substituted into equation 8 to derive $X_{t-1}$.Appendix D details the DDMSL model's process.
>
> It is important to note that during the training and testing of DDMSL, Gumbel Softmax is utilized to sample $X_{t-1}$ after obtaining its distribution (shown in lines 10 and 7 of Algorithm1 and Algorithm2 in Appendix D, respectively). Therefore, the inferred results of DDMSL at each time step are sampled from a probability distribution, and the state distribution of the next time step depends on the inferred results from the previous time step. As a result, DDMSL can generate a series of diverse source nodes and diffusion paths that conform to different diffusion patterns.
>
> (3)"the efficiency issue of the proposed framework."
>
> DDMSL uses a residual graph convolutional network as its backbone. For a graph with N nodes and E edges, the complexity of the network during inference from time t=T to t=0 is roughly $O(KT|E|)$(K is the number of GCN layers).
>
> During experiments, we found that the generation of $Q_t$ has the most significant impact on time complexity. $Q_t$ is derived by performing m SIR Monte Carlo simulations to calculate $Q_t$, resulting in a time complexity of approximately $O(mTN^2)$(m is the number of Monte Carlo simulations). This calculation can be time-consuming, especially for large-scale graphs.
>
> To reduce inference time in DDMSL, we employ the following strategies:
>
> 1)To optimize the computation of $Q_t$ using Monte Carlo simulation, we support code optimization for parallel computation. Our code enables accelerated computation through multi-threading. By utilizing CUDA operators, we can reduce inference time by 20-30 times (GPU version still undergoing testing). We also provide a comparison of the training and testing times of DDMSL using 12 threads alongside other benchmark algorithms (Due to time constraints, we used default settings for baseline algorithms.) in Table 3, where DDMSL demonstrates acceptable inference speed. Please refer to the attached PDF.
>
> 2)An alternative approach involves leveraging existing deep learning models(see Appendix D) to fit $Q_t$, requiring only a portion of the generated data for accurate predictions. This substitution can roughly reduce testing and training times by one-third.
>
> (4)"Limitations"
>
> We believe that the negative impact of our research on society is limited. One potential concern regarding DDMSL is its impact on privacy when reasoning about source nodes or diffusion paths. This issue may lead to the unintended exposure of sensitive information associated with certain nodes. For instance, in the context of an AIDS transmission network, patients who wish to keep their AIDS diagnosis confidential could potentially be identified. We acknowledge this concern and will address it in the upcoming revised version of our paper.
>
> Once again, we express  gratitude for your thorough and meticulous review, as well as for the valuable suggestions you have provided for our paper. We hope that our explanation adequately addresses your inquiries.

---

### Official Review · Reviewer_W7Kc · 2023-07-06

**Soundness:** 2 fair
**Presentation:** 2 fair
**Contribution:** 2 fair
**Rating:** 5
**Confidence:** 3

**Summary:**

This paper proposes a discrete denoising diffusion model for source localization called DDMSL. DDMSL can simultaneously locate information sources and restore information propagation paths. Experiments results on real-world datasets demonstrate DDMSL’s effectiveness.

**Strengths:**

1. This is the first study to solve information spread problems based on denoising diffusion model.
2. Extensive experiments on real-world datasets demonstrate the effectiveness of the proposed method.


**Weaknesses:**

1. The motivation of employing denoising diffusion model is not clear. Why is this generative model used for source localization problem? Compared to other source localization models, what are the advantages of the denoising diffusion model?
2. I do not agree that the proposed method is a denoising diffusion model. The final distribution of the forward process and the initial distribution of the reverse process are both random because they are simulated by the SIR model. It is more like a sequence model trained by the SIR data.
3. The motivation behind the model design is not clear. For example, why are SN and BN employed in Eq. 16?


**Questions:**

1. The phrase “graph inverse problem” or “inverse problem” sounds peculiar, perhaps the diffusion inverse problem or the spread inverse problem would be more comprehensible.
2. The formulation of the proposed denoising diffusion model is not clear. In Eqs. (4) and (5), why is $x^i_t$ included in $q(x^i_t | x^i_{t-1})$ and $q(x^i_t | x^i_0)$? If we already know $x^i_t$, is it necessary to have $x^i_{t-1} or x^i_0$ to calculate $x^i_t$?
3. How to obtain the distribution $Cat(x^i_t; p)$?
4. What are reaction diffusion models? Maybe they should be included in the Section Related Work?


**Limitations:**

1. The proposed model requires simulation of SIR from t_0 to t_T to obtain the training data, whereas diffusion models can be trained at any step by sampling Gaussian noise at an indicated time step. It seems that the model proposed in this paper needs more training time and is difficult to apply to large-scale graphs, such as OGB datasets. It is also necessary to compare time complexity with baseline methods.

---

> ### Author Rebuttal · Authors · 2023-08-08
>
> Thank you sincerely for your valuable suggestion. Taking into account your feedback, we have incorporated a comprehensive time complexity evaluation for DDMSL and will ensure that all your questions are answered thoroughly.
>
> (1) "What are the motivations and advantages of using diffusion models to handle inverse problems?"
>
> Please refer to our global response.
>
> (2) "The relationship between DDMSL and denoising diffusion model."
>
> As expounded in global response, we posit that the SIR model's role in the information diffusion process can be perceived as noise, progressively obfuscating the information contained within $X_0$. It becomes challenging to disentangle the inherent noise of the original $X_0$ data from the final input data utilized by DDMSL. In contrast to conventional denoising diffusion models like DDIM, which handle a sequence of artificially added Gaussian noise, our model processes a sequence of SIR data wherein the noise originates naturally. This fundamental distinction sets apart the noise characteristics encountered in SIR sequence data from those artificially introduced in DDIM.
>
> (3) "The motivation behind the model design is not clear. "
>
> The rationale behind the design of $nn_{\theta}$ stems from our belief that the propagation process of SIR resembles the message transmission process depicted in equation 14, thus enabling simplification to the form presented in equation 15. We will furnish a specific design case (termed as Equation 16) based on equation 15. Employing SN(·) and BN(·), we aim to constrain the Lipschitz coefficients of the model, as discussed in the proof of Lemma 3.1 (page 6, line 209), with additional detailed proof provided in the appendix. We sincerely apologize for any inconvenience caused to the readers. To enhance clarity, we will augment this section with more comprehensive explanations in the next revised version.
>
> (4) The use of the phrase “graph inverse problem”
>
> Diffusion occurs in various spatial and entity relationships. Our research focuses on graph-based analysis, hence the name is set to align with our investigation. We highly appreciate your input and will diligently take your opinion into consideration.
>
> (5) "The formulation of the proposed denoising diffusion model is not clear. "
>
> Equations 4 and 5 are crucial for modeling the information diffusion process in DDMSL. Eq.4 captures the relationship between node states at adjacent time steps, while Eq. 5 establishes the connection between $X_0$ and node states at any given time using a Markov chain derived from Eq. 4.
>
> To calculate our target $q(X_0 | X_t)$, we need to obtain $q(X_t | X_0)$ and establish the relationship between $X_t$ and $X_0$. However, we can only observe $X_t$ in reality, which corresponds to the infection data. Therefore, we must infer $X_{t-1}$ from $X_t$ iteratively until we obtain the inferred result of $X_0$.
>
> Various methods exist for inferring $X_{t-1}$ from $X_t$. For example, DDIM predicts the noise between $X_t$ and $X_{t-1}$ and subtracts it from $X_t$ to obtain $X_{t-1}$. In our case, we use a discrete denoising diffusion model (D3PM), which utilizes a neural network model to predict an intermediate $X_0^{'}$ (different from $X_0$) based on $X_t$. Then, $X_0^{'}$ is substituted into equation 8 to derive $X_{t-1}$. Appendix D details the DDMSL model's process.
>
> (6)"How to obtain the distribution $Cat(x_t^{i};p)$?"
>
> Let's illustrate the calculation process of Equation (4) with an example. At time $t=s$, node A has a state represented by $x_s=[1,0,0]$. The state transition matrix $Q_{s+1}$ at time $t=s+1$, computed using equations 2 and 3, is assumed to be $Q_{s+1}=[0.4,0.6,0; 0,0.7,0.3; 0,0,1]$. Thus, the state distribution of node A at time $t=s+1$ is $x_{s+1} = x_sQ_{s+1} = [0.4,0.6,0]$.
> This implies that node A is in state S  at time $t=s$, and the probability of it being in state S at time $t=s+1$ is:
>
>  $q(x_{s+1}=S|x_s=S)=x_sQ_{s+1}x_{s+1}^T=[0.4,0.6,0][1,0,0]^T=0.4$.
>
> Similarly, the probability of node A being in state I is 0.6.
>
> (7)"What are reaction diffusion models?"
>
> Our sincere apologies for any lack of clarity in explaining the reaction diffusion models. In the paper, "reaction diffusion models" refer to models like SIR, SI, etc., where nodes undergo reactions with neighboring nodes to update their states and influence others. We will enhance the introduction and related work sections to clarify these concepts.
>
> (8)"Limitations"
>
> DDMSL uses a residual graph convolutional network as its backbone. For a graph with N nodes and E edges, the complexity of the network during inference from time t=T to t=0 is roughly $O(KT|E|)$(K is the number of GCN layers).
>
> During  experiments, we found that the generation of $Q_t$ has the most significant impact on time complexity. $Q_t$ is derived by performing m SIR Monte Carlo simulations to calculate $Q_t$, resulting in a time complexity of approximately $O(mTN^2)$(m is the number of Monte Carlo simulations). This calculation can be time-consuming, especially for large-scale graphs.
>
> To reduce inference time in DDMSL, we employ the following strategies:
>
> 1) To optimize the computation of $Q_t$ using Monte Carlo simulation, we support code optimization for parallel computation. Our code enables accelerated computation through multi-threading. By utilizing CUDA operators, we can reduce inference time by 20-30 times (GPU version still undergoing testing). We also provide a comparison of the training and testing times of DDMSL using 12 threads alongside other benchmark algorithms (Due to time constraints, we used default settings for baseline algorithms) in Table 3, where DDMSL demonstrates acceptable inference speed. Please refer to the attached PDF.
>
> 2) An alternative approach involves leveraging existing deep learning models (see Appendix D) to fit $Q_t$, requiring only a portion of the generated data for accurate predictions. This substitution can roughly reduce testing and training times by one-third.

---

### Official Review · Reviewer_wXsK · 2023-07-07

**Soundness:** 3 good
**Presentation:** 3 good
**Contribution:** 2 fair
**Rating:** 5
**Confidence:** 5

**Summary:**

Information diffusion is common in various domains, such as social networks, the internet, and disease propagation. Obtaining the diffusion paths and localizing the source based on the final diffusion node states is beneficial for researchers to identify key transmission pathways during information dissemination and facilitate control over the propagation. This article proposes a framework for information diffusion called Discrete Diffusion Model for Source Localization (DDMSL) using the Susceptible-Infected-Recovered (SIR) model. The framework utilizes a discrete Markov Chain to characterize information diffusion and recover diffusion paths based on observed results while achieving source localization. For the neural network module in the framework, the authors design an inference model based on a residual network and rigorously prove its reversibility to justify the rationality of the model design. The experiments are performed on five real-world networks, where propagation data is generated using SIR and SI models. The experiment results demonstrate that, compared to the baseline, DDMSL performs better in source localization and diffusion path recovery across all datasets.

**Strengths:**

- Interesting idea. This paper proposes a method to establish a correlation between the information propagation process and the denoising diffusion model, which is worth further research.
- Rigorous theoretical guarantees. This paper presents reliable theoretical proofs to explain the designs of the model, which are also proven effective in the experiment stage.
- Well performance on different networks(datasets). The experiment results demonstrate the consistent performance across different networks and the improvement compared to the baselines.

**Weaknesses:**

- This paper needs to clarify why the diffusion model is being used for this task. Only the achievements of the diffusion model in the image inverse problems are mentioned. Personally, it seems that there is a significant difference between the inverse problems of images and the task addressed in this paper.
- In my view, the method proposed is not highly correlated to the so-called denoising diffusion model, as the initial input is not a particular noise, and the diffusion steps $T$ are set equal to the steps of information propagation.
- The paper does not show the model's generalization ability on real-world propagation datasets (only SIR or SI model as presented in the paper, no real-world datasets like *Digg* and *Memetracker*).
- Compared to previous works, the proposed method requires fine-grained diffusion history information, which may largely limit its application scenarios.
- Compared to previous works that do not explicitly simulate every diffusion step, such as SL-VAE, this method seems much more inefficient. And no efficiency study is provided.

## After Rebuttal
The authors' response has addressed most of my concerns. I suggest authors discuss the following points in the revised versions.
1. How to decide the number of diffusion/denoising steps in the diffusion model, the same as the real-world forward diffusion process on networks? What if this number is extremely large?

2. Please provide generalization performance on other real-world large networks, and comparison with baseline methods.

3. Requiring fine-grained diffusion history information is still a large limitation. Can this forward diffusion history be inferred with parameter estimation?

**Questions:**

My most concerning questions are mostly mentioned in the *Weaknesses* part, which are:

1. The reason why the proposed method utilizes the denoising diffusion model for this task.
2. The performance of the proposed method on real-world propagation datasets is not provided.
3. In IVGD(Wang et al.), the invertibility of the network is proven so that fixed-point iterations can be performed. I am concerned about how the reversibility of residual blocks, strictly proven, contributes to the prediction of $X_0$.(In Figure 2, the input is $Y_T$, while in the text of sec.3.4, the input is $X_0$.)
4. Moreover, I would like to know whether the model has been tested on different graph topologies under a single training. Because in most application scenarios, the trained model is asked to generalize on the changing graphs. I wonder whether the results in the paper are from the model trained and tested on the same static network topology(dataset).

**Limitations:**

The authors have illustrated the main limitation that the model requires prior knowledge about propagation models.

The provided source code cannot be run directly.

---

> ### Author Rebuttal · Authors · 2023-08-08
>
> Thank you for your valuable feedback. In response to your input, we have conducted experiments utilizing real-world diffusion datasets and DDMSL generalization experiments. The forthcoming section will provide comprehensive responses to each of your inquiries.
>
> (1)"Why the proposed method utilizes the denoising diffusion model for this task?"
>
> Response: Please refer to our global response.
>
> (2)"Relationship between the proposed method and the denoising diffusion model."
>
> Response: As mentioned in global response, our approach utilizes a discrete diffusion denoising model that is extensively employed in graph data generation tasks [2,3]. Unlike Gaussian distributed noise sets, this model deals with noisy data exhibiting a stable distribution resulting from the initial data's interaction with the state transition matrix. As illustrated in the paper, the SIR, SI, and other models can be effectively represented using state transition matrices. Since the diffusion noise sequence data is naturally generated, there is no need to introduce additional noise for constructing the noise sequence.
>
> Furthermore, Theorem 3.1 provides assurance regarding the convergence of node states at every moment, ensuring the model's functionality at any given time. Consequently, we set the time step to T in our experimentation.
>
> (3) "Showing the model's generalization ability on real-world propagation datasets."
>
> Response: Thank you for your suggestion. We tested DDMSL using real-world information dissemination datasets. However, the Digg dataset lacked propagation information flows due to recent updates. The Memetracker dataset's significant growth posed processing challenges within limited time. Thus, we used alternative datasets: Twitter [4] (12,627 nodes, 309,631 edges) and Douban [4] (23,123 nodes, 348,280 edges). These datasets underwent SLVAE benchmark algorithm processing, followed by source localization testing. Results can be found in Table 4 in the attached PDF.
>
> (4) "How the reversibility of residual blocks, strictly proven, contributes to the prediction of $X_0$ ?"
>
> Response: We will first explain the relationship between Section 3.4 and Figure 2, and then briefly introduce the process of DDSML and the roles of each module.
>
> a. The relationship between Section 3.4 and Figure 2.
>
> Figure 2 illustrates a reversible residual network used in DDMSL. This network functions similarly to Unet or Transformer in an image diffusion model (DDIM). Please refer to Appendix D for detailed training and inference processes. The network takes observed infection information ($Y_T$) as input and outputs the potential source node ($X'_0$) that might have caused $Y_T$.
>
> Moreover, Section 3.4 primarily focuses on constructing a neural network model capable of fitting the SIR inverse diffusion process, which is introduced in Section 3.4 on page 5. Equation 13 demonstrates that assuming $X_0$ represents the source nodes, we utilize $Y_T=P(X_0)$ to represent the generation of $Y_T$ from $X_0$ using the SIR or SI process. Naturally, $X_0$ is unknown while $Y_T$ is known, and inferring $X_0$  from $Y_T$ can be denoted as $X_0=P^{-1}(Y_T)$.
>
> Lastly, we devised a residual block based on GCN, built upon the equivalence between SIR/SI diffusion and message passing processes, and assembled these blocks to form a reversible network. Subsequently, we verified that the network is indeed reversible. Our ultimate objective is to design the residual network depicted in Fig2, denoted as $P^{-1}$, ensuring its capability to output $X_0$ after training with $Y_T$ as input.
>
> b. How the reversibility of residual blocks, strictly proven, contributes to the prediction of $X_0$?
>
> Now let's delve into this matter further. The group of residual blocks forms a network that improves the receptive field of GCN and addresses issues like node oversmoothing. This network corresponds to fitting $P^{-1}$. We input the observed infection graph $Y_T$ into the residual network, calculate the loss between the output and the actual $X_0$, and update the network's parameters. Through iterations, the residual network gradually approximates the true $P^{-1}$. Rigorous theoretical proofs assure that residual networks can produce $X_0$ as an output.
>
> (5) "Whether the model has been tested on different graph topologies under a single training?"
>
> Response: Our experiments entail separate training and testing processes for each graph structure, following the prevailing methodology in deep learning algorithms for source localization.
>
> We wholeheartedly concur with your viewpoint regarding the significance of evaluating model generalization ability. To evaluate generalization, we conducted additional experiments using three diverse network datasets of varying scales. Each trained model was assessed on different network structures, revealing the DDMSL model's remarkable generalization performance. Detailed outcomes are provided in Table 2 of the attached PDF.
>
> (6) "Limitations"
>
> Response: As elucidated in our paper, DDMSL does possess limitations that necessitate prior knowledge of partial diffusion models. However, these limitations are not insurmountable, given the availability of alternative approaches to aid in parameter estimation for information diffusion models (see Sec 5). In addition, We have made updates to the code.
>
> [1] Song, Jiaming ,  C. Meng , and  S. Ermon . "Denoising Diffusion Implicit Models." (2020).
>
> [2] Vignac, Clement, et al. "Digress: Discrete denoising diffusion for graph generation." arXiv preprint arXiv:2209.14734 (2022).
>
> [3] Haefeli, Kilian Konstantin, et al. "Diffusion models for graphs benefit from discrete state spaces." arXiv preprint arXiv:2210.01549 (2022).
>
> [4] Z. Cao, K. Han and J. Zhu, "Information Diffusion Prediction via Dynamic Graph Neural Networks," 2021 IEEE 24th International Conference on Computer Supported Cooperative Work in Design (CSCWD), Dalian, China, 2021, pp. 1099-1104, doi: 10.1109/CSCWD49262.2021.9437653.

---

> > ### Comment · Reviewer_wXsK · 2023-08-21
> > **Thanks for your response**
> >
> > Thanks for your detailed response. However, I still think the following weaknesses hinder the applicability of DDMSL.
> > 1. The number of diffusion steps is fixed as the real-world forward diffusion process on networks. What if this number is extremely large?
> >
> > 2. Authors did not provide generalization performance on other real-world large networks (attached PDF, Table 2), and comparison with baseline methods.
> >
> > 3.  Requiring fine-grained diffusion history information is still a large limitation. Can this forward diffusion history be inferred with parameter estimation?

---

> > > ### Author Response · Authors · 2023-08-21
> > > **Some clarifications on DDMSL generalization performance and limitations.**
> > >
> > > Thank you for raising these important questions. We have carefully considered these issues and would like to share our thoughts on them.
> > >
> > > 1. The number of diffusion steps is fixed as the real-world forward diffusion process on networks. What if this number is extremely large?
> > >
> > > Response: First, we propose that the size of information diffusion is the primary factor influencing the performance of source location, rather than the number of diffusion steps. Taking the SI model as an example, when the diffusion step size is sufficiently large, almost all nodes will be in the infected state, making it impossible to identify the source node using any algorithm. In the context of predetermined initial parameters for the entire diffusion system, the diffusion scale increases as the diffusion step size increases. The impact of diffusion scale on source location performance has been discussed in previous research [1].
> > >
> > > Secondly, extremely large diffusion steps are rarely observed in real-world scenarios since they imply complete information diffusion throughout the network. Currently, no algorithm is capable of handling such a situation. Hence, the majority of source location methods, including DDMSL, regulate the diffusion step during the initial and intermediate stages of information dissemination.
> > >
> > > Therefore, in DDMSL, we opt to locate the source based on a specific diffusion scale (e.g., when 50% of nodes are infected). However, achieving this specific diffusion scale requires different numbers of diffusion steps depending on the set of initial diffusion parameters. Our experiments align with the aforementioned statement: source detection performance is mainly influenced by the diffusion scale. Moreover, DDMSL exhibits similar performance when detecting infection patterns with different diffusion steps but the same diffusion scale.
> > >
> > > [1] Shah, Chintan , et al. "Finding Patient Zero: Learning Contagion Source with Graph Neural Networks." (2020).
> > >
> > > 2. Authors did not provide generalization performance on other real-world large networks (attached PDF, Table 2), and comparison with baseline methods.
> > >
> > > Response: We appreciate your insightful comments. Indeed, the generalization performance of the source location algorithm is of great importance. However, it is worth noting that, to the best of our knowledge, previous studies have not specifically examined this aspect, which is why in our original submission we did not include a comparison with the generalization performance of the baseline algorithm in our evaluation of DDMSL.
> > >
> > > The experimental results in this paper are based on synthetic data sets on real networks with different sizes and structures. The results presented in Table 2 in the attachment demonstrate that DDMSL exhibits strong performance in both small world and BA network structures. Although there is a notable decline in performance in extreme cases such as ER networks, the results remain acceptable. These findings highlight the robust generalization capabilities of the DDMSL algorithm. We believe that DDMSL should also perform well on two extensively diffuse datasets.  However, due to time constraints during the discussion phase, we may not be able to provide the results of subsequent supplementary generalization experiments in a timely manner. If necessary, we will include these results in our next revision to further support our findings.
> > >
> > > 3. Requiring fine-grained diffusion history information is still a large limitation. Can this forward diffusion history be inferred with parameter estimation?
> > >
> > > Response: In fact, taking the main comparison algorithm SLVAE as an example, DDMSL only requires additional diffusion information during the inference process, specifically the initial time diffusion parameters (such as infection rate and recovery rate).
> > >
> > > During the training process, obtaining diffusion history data in most real-world scenarios is straightforward. Even if such data is not available, it is still possible to synthetically generate data of the same diffusion scale for training purposes after estimating the initial diffusion parameters. DDMSL is trained and tested on the real diffusion dataset using this method, that is, the training set is synthesized after the parameters are evaluated, and finally tested on the real dataset. Table 4 in the attachment shows the feasibility of this method.
> > >
> > > Consequently, this limitation does not pose a significant concern.

---

### Official Review · Reviewer_NqvF · 2023-07-09

**Soundness:** 2 fair
**Presentation:** 1 poor
**Contribution:** 3 good
**Rating:** 6
**Confidence:** 4

**Summary:**

This paper addresses the problem of source identification in a stochastic
network diffusion process, having observed the set of nodes that are
infected at time T. It considers SIR and SI models. The authors propose a
neural network based solution called DDMSL. The authors approach is based
on a reaction-diffusion process and formulate it using a message-passing
function and use a reversible residual network for source identification
and reconstruction of the cascade. DDMSL is compared with several
baselines and an ablation study is conducted to evaluate the significance of
the reversible residual network and propagation rule supervision module.

**Strengths:**

The authors use a novel reversible residual network construction for the
source identification problem.

The paper has a strong experiments component considering multiple networks
and several baselines.

DDMSL shows much superior performance compared to the baselines.

**Weaknesses:**

The presentation could be greatly improved. Many notation and crucial
concepts are missing. The derivations are very dense and lack intuitive
explanations. So, it is hard to understand the results. Details follow.

Diffusion models are not clearly defined: There are several types of SIR
and SI models in the literature. The authors fail to describe their model
clearly. Is it a discrete-time SIR process, i.e., an infected node infects
its neighbors with probability equal to the edge weight at each time step
until it recovers, which again is determined by some probability? A second
type of process is a Gillespie process which is a discrete-event process
which involves rates as the authors have used. Also, the authors seem to
use a single transmission rate per node. This is important to clarify as
the baselines considered in the paper use their own definitions of SIR
models. Is it consistent with the definition of the SIR model in this
paper? If not, then the comparison wouldn't be fair.

Reaction-diffusion process is not defined. In section 3.4.1, the authors
seem to relate GCN with reaction-diffusion models. This is not clear at all
as the concept is not introduced.

Theorem 3.1: What is the purpose of this theorem? Further, its proof is not
presented, even in the appendix. Also, it is not referred to anywhere.
Equation (3) is not used anywhere as well. It is not clear why the rates
can be expressed in that manner. Also, even though these are mentioned as
rates, beta and gamma are used as probabilities.

Several symbols are not defined: nn_\theta, KL, D_KL, Cat(), Q_k^i. It is
very difficult to follow the derivations. They are dense and lack
explanations.

Ablation: In DDMSL(a), the reverse residual block is removed, but replaced
by a GCN network. Typically, in an ablation study, the component is removed
and the performance evaluated. It is possible that the GCN network
deteriorates the performance.

The Figures 6 to 11 in the Appendix are very difficult to read and infer.
Besides, these are just few instances where DDMSL does well. Table 2
already quantifies the performance.

**Questions:**

Several definitions of notation are missing. Diffusion model is not
defined. These are mentioned in the Weakness section.

In Algorithm 1 line 4, t is sampled from a distribution where the
probability of t increases as its value increases. This is not explained
anywhere. Why is this sampling necessary?

**Limitations:**

No negative societal impact.

---

> ### Author Rebuttal · Authors · 2023-08-08
>
>    Thank you very much for your comments and suggestions. We have provided some explanations and clarification regarding your concerns as follows:
>
> (1) “Many notation and crucial concepts are missing”.
>
>    Response: Thank you very much for your comments and suggestions.  The neural network model used in the reverse inference process was described in section 3.3.2 on page 4 by ${nn_{\theta}}$.
> We have only used $Q_t^i$ instead of $Q_k^i$ in our manuscript, and specific explanations of $Q_t^i$ are provided on page 4, line 121. As for $KL, D_ {KL}, Cat()$, these symbols are from the Loss function of the Diffusion model, which have been used in many diffusion models widely, such as DDIM[3]. We deeply apologize for any confusion caused to readers. We will provide a more detailed introduction to these symbols in subsequent revised version.
>
> As for the lack of the concept of reaction diffusion process, in our paper, reaction diffusion process refers to individual mediated diffusion processes such as SIR (see equation 1), SI, SIS, etc. We will revise and add this section in subsequent paper revisions. Thank you for your suggestion.
>
> (2) “Diffusion models are not clearly defined”.
>
> Response: We use discrete SI and SIR models, please refer to equation 1 on page 3 for detailed definitions. During the experiment, all baseline algorithms used consistent SIR and SI models and ensured that the experimental results were generated based on the same datasets.
> In fact, the transition probability of each node in the SIR process is necessarily different, just as Lokhov et al. [1] analyzed the SIR process using message passing algorithms, and we inferred the state distribution of any node at any time step. Each node follows the pre-defined process throughout the entire diffusion process: an affected node affects its neighbors with probability equal to the edge weight at each time step until it recovers. However, the infection probability and recovery probability of each node during this period depend on the state of the node and its neighbors, as SIR is a Markov process.
>
> (3)“The proof of Theorem 3.1 and the role of  Equation 3”.
>
> Response: The purpose of Theorem 3.1 is to ensure that the state distribution of nodes at each time step is convergent, which is a necessary condition for the discrete Diffusion model to work [2]. Due to space limitations, we have provided a detailed explanation of its proof and function in the appendix. Please refer to section C.1 on page 16 for details. Therefore, our proof of Theorem 3.1 is not missing. And, we will add a description of Theorem 3.1 in the main text.
>
> Equation (3) is not that what you said has not been used anywhere. If you read the paper carefully, the role of Equation (3) is explained in lines 121-124 on page 4, that is, the parameters used to calculate the State-transition matrix of Equation (2). Meanwhile, the generation of Equation (3) is also very intuitive. For instance, the probability of node $i$ being infected at time $t+1$ is the sum of the two: the probability of $i$ being uninfected and subsequently infected at time t, and the probability of $i$ being infected at time $t$ but not transitioning to a recovery state afterwards. We will add a detailed explanation of this section in the appendix.
>
> (4) “Ablation Study“.
>
> Response: Thanks for the great suggestion. We have added ablation experiments on the GCN module, and the experimental results are shown in Table 1 and Figure 1 in the attached PDF, where DDMSL (c) represents the model after removing the GCN module.
>
> (5) “The Figures 5 to 11 in the Appendix“.
>
> Response: Figures 5 to 9 are visualizations of DDMSL in the process of reconstructing information diffusion, while Figures 10 to 11 are visualizations of source localization tasks. Each graph has thousands or even tens of thousands of nodes, and it is impossible to display the reconstruction process of each dataset in such a small space. Therefore, we choose to use vector graphs to clearly see the state of each node and even each edge when zoomed in, but this will result in a significant memory footprint.
>
> We believe that visualization of reconstruction diffusion is necessary, and the source localization results can already be clearly displayed in Table 2. Therefore, we have reduced the visualization of source localization to prevent further expansion of text memory usage.
>
> (6) “Reason for sampling t in Algorithm 1 line 4“
>
> Response: The probability of t being sampled is indeed positively correlated with  $t$. Generally, $t$ is uniformly sampled in the Diffusion model of other fields. But we found in the experiment that, compared to the uniform sampling, our setting can significantly reduce the training time and bring some performance improvement. Sorry for missing the explanation on t sampling, and we will supplement it in subsequent modifications.
>
> (7) “Limitations“.
>
> Response: We believe that the negative impact of our research on society is limited. It is possible that some people's privacy may be exposed in some use scenarios, e.g., AIDS patients who do not want to disclose their diseases. In summary, we will also add this section in the appendix.
>
> [1] Lokhov, Andrey Y., Mézard, Marc, and Zdeborová, Lenka. "Dynamic message-passing equations for models with unidirectional dynamics." (2014).
>
> [2] Austin, Jacob, et al. "Structured Denoising Diffusion Models in Discrete State-Spaces." (2021).
>
> [3]Song, Jiaming ,  C. Meng , and  S. Ermon . "Denoising Diffusion Implicit Models." (2020).

---

> > ### Comment · Reviewer_NqvF · 2023-08-19
> >
> > I am satisfied with the answers. I do not have any further queries.

---

### Author Rebuttal · Authors · 2023-08-08

I would like to extend my sincere appreciation to the esteemed reviewers for their invaluable suggestions on our paper. It has come to our attention that several reviewers have expressed interest in understanding the rationale behind our adoption of the denoising diffusion model. We will address this and share additional experiments conducted based on reviewer feedback, including testing DDMSL on real-world diffusion data, generalization experiments, time complexity analysis, and supplementary ablation experiments. Please refer to the attached PDF file for the experimental results.

We will introduce our motivation from the following aspects:

a. "Differences between discrete diffusion denoising models and classical diffusion models (such as DDIM)"

Firstly, it is important to clarify that we are employing the D3PM [2] diffusion denoising model in a discrete space, which distinguishes it from classical diffusion models like DDIM [1]. D3PM leverages Markov chains to model discrete data, where the data between two consecutive time steps are interconnected through the state transition matrix $Q_t$. In contrast to DDIM, which introduces Gaussian noise directly to the data to establish a forward process, D3PM incorporates noise into $Q_t$ to account for the stochastic nature of the diffusion process.

b. "Comparison of SIR diffusion and noise diffusion on images"

The application process of diffusion models in image analysis can be summarized as follows: starting with an original image, noise is continuously introduced to approximate a Gaussian distribution. DDIM, in particular, performs continuous denoising on the Gaussian distribution noise to generate a new image. In information dissemination models such as SIR , the process of information diffusion bears similarities to the noise introduction process in image analysis. Given a set of initial seed nodes $x_0$, their states gradually become perturbed under the influence of SIR rules. Subsequently, the noise imposed by SIR rules is eliminated from the final noisy infection graph. We hypothesize that the state of $x_0$ evolves over time based on the impact of SIR rules, consequently causing changes in the states of $x_0$'s neighbors and even higher-order neighbors. This progressive alteration obscures the original information encoded in $x_0$. From this perspective, the node state modification at each time step induced by SIR can be interpreted as noise introduced during the evolution of information diffusion.

Models like SIR,SIS and SI epitomize the most natural diffusion processes.	For instance, in the SIS model, which can also be processed b DDMSL, when specific network topology conditions are met, and given a set of seed nodes, as the propagation time increases, each node eventually converges to a stable distribution where the probabilities of being in the susceptible(S) and infected(I) states approach 0.5. In fact, Theorem 3.1 demonstrates that the node state distribution of both the SI  and SIR  models at each time step during the diffusion process exhibits convergence.

Hence, it is our contention that the dissemination of information on a graph and the propagation of noise in an image are two distinct occurrences transpiring in discrete and continuous spaces, respectively. This observation serves as one of the motivations behind employing diffusion models for source localization. As previously discussed, we leverage the D3PM to address this task. The SIR, SI, and SIS models align seamlessly with the utilization requirements of D3PM, and the formulation of the state transition matrix (as specified in Equation 2) for each node is straightforward.

c. "Why use generative models for source localization?"

In the context of an observed infection graph ($Y_T$), numerous sets of source nodes ($x_0$) can exist, as depicted in Fig1 of the paper. Inferring the distribution of source nodes using posterior diffusion data constitutes a critical task. However, existing source localization algorithms such as LPSI, OJC, and other methods employed in Baseline generate deterministic and imprecise outcomes. This limitation motivates the utilization of VAE in SLVAE. In fact, this forms the foundation of DDMSL and SLVAE's approach to source localization.

d. "Why use diffusion models for source localization? What are their advantages?"

Firstly, the diffusion model demonstrates superior performance compared to traditional generative models like VAE in terms of task generation accuracy. In our experimental evaluation, DDMSL outperformed generative models such as SLVAE and DDMIX for source localization purposes.

Secondly, the unique advantage of the diffusion model lies in its ability to leverage the inference results from previous time steps to inform the inference process at each subsequent time step. As a result, it can construct comprehensive node information at each time step, aligning perfectly with our objective of reconstructing the communication path. This aspect serves as an additional motivation for adopting the discrete diffusion model for our task.

In summary, the reasons for using the discrete diffusion model are as follows:

1) The phenomenon of information diffusion is a pervasive occurrence in various domains, making the adoption of D3PM suitable for modeling forward processes and learning reverse processes.
2) DDMSL exhibits remarkable performance in reconstructing the states of individual nodes at each time step throughout the propagation process, enabling successful accomplishment of the two primary tasks outlined in the paper: source localization and path reconstruction.
3) When compared to alternative algorithms like SLVAE and DDMIX, which also employ generative approaches for source localization, DDMSL demonstrates superior performance.

[1] Song, Jiaming ,  C. Meng , and  S. Ermon . "Denoising Diffusion Implicit Models." (2020).

[2] Austin, Jacob , et al. "Structured Denoising Diffusion Models in Discrete State-Spaces." (2021).

---

### Decision · Program_Chairs · 2023-09-21

**Decision:**

Accept (poster)

**Comment:**

The paper introduces a novel reversible residual network construction for source identification. Extensive experiments involving multiple networks and baselines demonstrate DDMSL's superior performance. This is a borderline paper. The original reviewer scores were quite mixed. After the rebuttal, some reviewers seemed to be satisfied with the authors' rebuttal and slightly raised the score. I hope in the final version, the authors can further justify the motivation to use generative model for sources localization and improve the overall presentation of this paper.